

**Water availability limits tree productivity, carbon stocks, and carbon residence time in**
**mature forests across the western United States**
Logan T. Berner *[1], Beverly E. Law [1], and Tara W. Hudiburg [2]
[1] Department of Forest Ecosystems and Society
Oregon State University
321 Richardson Hall
Corvallis, Oregon 97331-2212
[2] Department of Forest, Rangeland, and Fire Sciences
University of Idaho
875 Perimeter Drive
Moscow, Idaho 83844-1133
* Corresponding Author:
logan.berner@oregonstate.edu
phone: 702-683-9987
fax: 541-737-5814
*Running header:*
Water limitations on forests in the western US
*Keywords:*
Ecological gradient, climate gradient, carbon cycle, climate change, climate moisture index,
forest inventory, MODIS, satellite remote sensing, biomass
*Type of paper:*
Primary research article



**Abstract**
Much of the western US is projected to become warmer and drier over the coming century,
underscoring the need to understand how climate influences terrestrial ecosystems in this region.
We quantified the response of tree net primary productivity (NPP), live biomass (BIO), and
mean carbon residence time (CRT=BIO/NPP) to spatial variation in climatic water availability in
the western US. We used forest inventory measurements from 1,953 mature stands ($\geq$100 years)
in Washington, Oregon, and California (WAORCA) along with satellite and climate data sets
covering the western US. We summarized forest structure and function in both domains along a
400 cm yr$^{-1}$ hydrologic gradient, quantified with a climate moisture index based on the difference
between precipitation and reference evapotranspiration summed from October-September (i.e.,
water-year) and then averaged annually from 1985-2014 ($CMI_{\overline{wy}}$). Median NPP, BIO, and CRT
computed at 10 cm yr$^{-1}$ intervals along the $CMI_{\overline{wy}}$ gradient increased monotonically with
increasing $CMI_{\overline{wy}}$ across both WAORCA ($r_s$=0.93-0.96, p<0.001) and the western US ($r_s$=0.93-
0.99, p<0.001). Field measurements from WAORCA showed that median NPP increased from
2.2 to 5.6 Mg C ha$^{-1}$ yr$^{-1}$ between the driest and wettest 5% of sites, while BIO increased from 26
to 281 Mg C ha$^{-1}$ and CRT increased from 11 to 49 years. The satellite data sets revealed similar
changes over the western US, though these data sets tended to plateau in the wettest areas,
suggesting that additional efforts are needed to better quantify NPP and BIO from satellites in
high-productivity, high-biomass forests. Our results indicate that mature forests in this region
were widely sensitive to changes in water availability, suggesting that projected climatic change
over the coming century could reduce NPP, BIO, and CRT in many parts of this region,
particularly the Southwest, with resulting impacts on ecosystem services.



## 1    Introduction


Climatic water availability strongly affects the distribution of plants on Earth's land surface
(Holdridge, 1947;Major, 1963) and the resulting structure and function of terrestrial ecosystems
(Schuur, 2003;Churkina and Running, 1998;Law et al., 2002). For instance, productivity of
desert (Whittaker and Niering, 1975), grassland (Yang et al., 2008) and forest (Schuur,
2003;Law et al., 2002;Berner and Law, 2015) ecosystems varies along spatial gradients in
climatic water availability. Climatic water availability is projected to change in many parts of the
world over the coming century in response to atmospheric warming from sustained
anthropogenic greenhouse gas emissions (Dai, 2013;Collins et al., 2013;Walsh et al., 2014).
Rising atmospheric temperatures increase evaporative demand (Hobbins et al., 2012) and the
probability that periods with anomalously low precipitation co-occur with anomalously high
temperatures, which increases the frequency and severity of drought (Diffenbaugh et al., 2015).
Societies depend on the goods and services provided by terrestrial ecosystems (e.g., forests;
Williams, 2006) and thus it is imperative to elucidate climatic controls over ecosystem structure
and function to help anticipate and mitigate potential impacts of ongoing climatic change.

Atmospheric warming in the western United States has increased the risk of drought and

continued warming over the coming century could reduce water availability in much of the
region (Diffenbaugh et al., 2015;Kunkel et al., 2013;Williams et al., 2012). Regional mean
annual temperatures increased 0.8-1.1°C from 1895 to 2011, while concomitant changes in
precipitation were more variable (Kunkel et al., 2013;Mote et al., 2014). Coincident with
regional warming there was an increase in area annually affected by drought (Cook et al., 2004)
and an increase in drought frequency (McCabe et al., 2004). The 2000-2004 drought was the
most severe drought to have occurred in the region during the past 800 years (Schwalm et al.,





2012). In 2013-2015, much of the western US experienced record low soil moisture and
mountain snowpack along with persistent high temperatures that exacerbated the multi-year
drought (Singh et al., 2016;Diffenbaugh et al., 2015). Climate models project mean annual
temperatures could further increase ~3.8-5.5°C by the end of the 21$^{st}$ century under a high
greenhouse gas emission scenario (RCP 8.5; Walsh et al., 2014;Kunkel et al., 2013). Models also
suggest that mean annual precipitation might increase ~10% in the northern part of the region,
though change little in the southern parts under the same high-emissions scenario; however,
these projections are more uncertain than projected changes in temperature (Walsh et al.,
2014;Kunkel et al., 2013). The recent severe droughts and projected increases in regional
atmospheric temperatures potentially foreshadow a shift towards hotter, drier conditions in much
of the region over the coming century (Collins et al., 2013;Dai, 2013;Williams et al.,
2012;Schwalm et al., 2012).
Changes in ecosystem structure and function along spatial climatic gradients can provide
insight into long-term ecosystem response to climatic change (Jin and Goulden, 2014;Biederman
et al., 2016;Berner et al., 2013). Mean annual precipitation varies over 500 cm yr$^{-1}$ across the
western US (Daly et al., 2008) contributing to a range of ecosystems from dry desert shrublands
to coastal temperate rainforests (Franklin and Dyrness, 1988;Waring and Franklin, 1979) where
live tree biomass (BIO) attains levels thought to be exceeded only be primary *Eucalyptus*
*regnans* forests in southern Australia (Keith et al., 2009;Waring and Franklin, 1979). Field
studies carried in this region found that BIO and/or net primary productivity (NPP) tended to
increase as conditions became wetter (Huxman et al., 2004;Knapp and Smith, 2001;Webb et al.,
1983;Berner and Law, 2015;Whittaker and Niering, 1975;Gholz, 1982); however, each study
was based on fewer than 20 field sites selected using a set of criteria (e.g., mature forest near a



road). Several of these earlier studies also indicated that mean carbon residence time
(CRT=BIO/NPP) in live aboveground biomass (AGB) increased several fold between the driest
and wettest plant communities (Webb et al., 1983;Whittaker and Niering, 1975;Gholz, 1982),
which is potentially related to differences in disturbance regimes and carbon allocation (Girardin
et al., 2010). These studies illustrate that ecosystem structure and function are strongly
influenced by water availability in parts of the western US; however, additional efforts are
needed to assess forest response to variation in water availability at larger scales across this
region.

Our objective in this study was to explore how forest structure and function changed

along spatial gradients in climatic water availability in the western US. We hypothesized that
tree NPP, BIO, and CRT in mature stands (>100 years old) are constrained by water availability
in this region. We thus anticipated that NPP, BIO, and CRT would increase as climate became
wetter (or, conversely, decrease as climate became drier). We tested these hypotheses first across
Washington, Oregon, and California (WAORCA) using forest inventory measurements from
1,953 sites and then across 18 Mha of mature forest in the western US using satellite remote
sensing data sets that included three national biomass maps and NPP derived from the Moderate
Resolution Imaging Spectroradiometer (MODIS). Tree NPP, BIO, and CRT were based on
above- and below-ground components. We quantified water availability using a climate moisture
index (CMI) that accounted for the cumulative difference between precipitation (P) and reference
evapotranspiration ($ET_0$) over the approximate seasonal cycle of soil water recharge and draw-
down (October-September; i.e., water year).





**2      Materials and methods**
**2.1      Data sets and preprocessing**
**2.1.1    Field estimates of tree biomass, productivity, and carbon residence time**
We using field measurements to estimate BIO ($BIO_{field}$; Mg C ha$^{-1}$), NPP ($NPP_{field}$; Mg C ha$^{-1}$ yr$^{-1}$
$^{1}$), and CRT ($CRT_{field}$; year) at 1,953 forest inventory sites located in mature stands spread across
WAORCA. These 1-ha sites were surveyed by the US Forest Service from 2001 to 2006 and
comprise a representative sample of forest lands in the region. We included sites in our analysis
when stand age was at least 100 years. Stand age was defined as the average age of the oldest
10% of trees, where individual tree age was determined on survey plots using increment cores
(Van Tuyl et al., 2005). $BIO_{field}$ and $NPP_{field}$ were computed for each site as part of a prior study
(Hudiburg et al., 2011). $BIO_{field}$ was estimated using regional allometric equations for tree
components (e.g., stem, branch, bark, foliage, and coarse roots) based on tree diameter and/or
height, along with estimates of fine root mass derived from a relationship with leaf area index
(LAI; m$^2$ leaf m$^{-2}$ ground). $NPP_{field}$ was estimated based on changes in above- and below-ground
woody biomass over a 10-year interval plus annual foliage and fine root turn-over. See Hudiburg
et al. (2011) for additional details. *Carbon residence time* is a key ecosystem characteristic that
describes the average duration, in years, that a carbon molecule will remain in a specific pool
(e.g., live biomass; Waring and Running, 2007). We computed $CRT_{field}$ in live tree biomass as
the ratio of $BIO_{field}$ to $NPP_{field}$ in stands averaging >100 years of age.

**2.1.2    Remote sensing estimates of tree biomass, productivity, and carbon residence time**



We used satellite remote sensing and ancillary data sets to estimate BIO (BIO$_{sat}$), NPP (NPP$_{sat}$),
and CRT (CRT$_{sat}$) across mature forests in the western US. BIO$_{sat}$ included the same component
carbon pools as BIO$_{field}$ (i.e, stem, branch, bark, foliage, coarse roots and fine roots). We
quantified the amount of carbon in stems, branches, and bark using an ensemble of three
satellite-derived data sets that depicted live tree aboveground biomass (AGB; excluded foliage)
circa 2000 to 2008 (Blackard et al., 2008;Wilson et al., 2013;Kellndorfer et al., 2012). Each map
was generated using satellite and geophysical (e.g., climate, topography) data sets to spatially
extrapolate forest inventory measurements over the conterminous US. We acquired these maps at
250-m spatial resolution and then converted two of the maps (Blackard et al., 2008;Kellndorfer
et al., 2012) from dry biomass to carbon assuming a 50% conversion factor (Smith et al., 2006).
We then reprojected these maps onto a uniform grid in an equal area projection, masked them to
the common forest extent, and then computed the ensemble average. We used the ensemble
average in the subsequent analysis given previous work showing that the ensemble average
agreed more closely with state-level estimates of tree aboveground carbon stocks derived from
forest inventories than any of the individual maps (Berner et al., in review).

After deriving spatial estimates of carbon storage in AGB, we then estimated carbon

storage in coarse roots, fine roots, and foliage for each 250-m forested pixel. As with AGB, we
assumed that roots and foliage were 50% carbon (Smith et al., 2006;Berner and Law, 2016). We
computed coarse root biomass based on an empirical relationship with AGB (Cairns et al., 1997)
and fine root biomass based on an empirical relationship with peak summer LAI (Van Tuyl et al.,
2005). Spatial estimates of LAI were available globally at 1-km resolution from NASA's
Moderate Resolution Imaging Spectroradiometer (MODIS) as part of the MOD15A2 (Collection
5) data set (Myneni et al., 2002). We obtained these LAI estimates at 8-day intervals during July



and August (late-summer) from 2000 to 2014 for the western US. We then (1) excluded poor-
quality pixels using the quality control flags; (2) computed average late-summer LAI over the
15-year period; and (3) reprojected and resampled the data set to the common 250-m resolution
equal area grid. We used average late-summer MODIS LAI to computed both fine root biomass
(as described above) and foliage biomass. Foliage biomass was estimated for each pixel by
dividing LAI by the average specific leaf area (SLA; g C m$^{-2}$ leaf) of the forest type found in that
pixel. We aggregated an existing map of forest type (Ruefenacht et al., 2008) into nine classes
(e.g., *Pinus ponderosa*, true fir) and then varied SLA among classes using species-, genus-, or
division-specific estimates of average SLA from a recent leaf trait synthesis (Berner and Law,
2016). We then estimated $BIO_{sat}$ for each 250-m resolution pixel by summing the above- and
below-ground carbon pools.

We quantified regional NPP using the satellite-derived MODIS primary productivity data

set ($NPP_{sat}$; MOD17A3 v. 55). The MODIS light-use efficiency model predicts global terrestrial
NPP each year at 1-km resolution by incorporating estimates of LAI, absorbed
photosynthetically active radiation, and land cover derived from MODIS with plant
physiological characteristics and climate data (Running et al., 2004;Zhao et al., 2010). The
model predicts annual NPP as the cumulative difference between daily gross primary
productivity and daily to annual plant respiration. These estimates thus reflect NPP allocated
both above- and below-ground. We obtained annual NPP estimates from 2000 to 2014 for the
western US, reprojected the data onto an equal area grid, and then averaged over years.

Several additional preprocessing steps were required after deriving forest $BIO_{sat}$ and

$NPP_{sat}$. These included masking both $BIO_{sat}$ and $NPP_{sat}$ to areas mapped as forest by the MODIS
land cover map (Friedl et al., 2010) and then further masking these data sets to include only areas





where stand age was at least 100 years. The map of stand age reflected conditions c. 2006 and
was produced by Pan et al. (2011) by combining forest inventory measurements, information on
historical fires, and optical satellite imagery. We applied these 1-km resolution masks to the 250-
m resolution $BIO_{sat}$ assuming homogenous land cover and stand age within each 1-km pixel. We
then average aggregated $BIO_{sat}$ from 250-m to 1-km resolution and computed $CRT_{sat}$ as the ratio
of $BIO_{sat}$ to $NPP_{sat}$.

### 196    2.1.3    Climate date sets and derivation of the climate moisture index

We quantified water availability using a climate moisture index (CMI) that was computed at
monthly time steps as precipitation minus $ET_0$ (Hogg, 1994;Webb et al., 1983). We summed
monthly CMI over each water-year (October in year *t-1* to September in year *t*) from 1985 to
2014 and then averaged over years to produce a 30-year climatology ($CMI_{\overline{wy}}$; cm $yr^{-1}$). The
water year represents the approximate annual cycle of soil water recharge and withdrawal
(Thomas et al., 2009). We obtained estimates of monthly precipitation from the Parameter-
elevation Relationships on Independent Slopes Model (PRISM; Daly et al., 2008), which
interpolated weather station measurements onto a 4-km resolution grid. We then estimated
monthly $ET_0$ using the Food and Agricultural Organizations (FAO) Penman-Monteith equation
(FAO-56; Allen et al., 1998), where
$$ET_0 = \frac{0.408\Delta\,(R_n - G) + \gamma\left(\frac{900}{T + 273}\right)U(e_s - e_a)}{\Delta + \gamma\,(1 + 0.34\,U)}$$



Variables included net incoming radiation ($R_n$), soil heat flux (G), mean daily temperature (T),
wind speed (U), and both saturation ($e_s$) and actual vapor pressure ($e_a$), as well as the
psychrometric constant ($\gamma$) and the slope of the vapor pressure curve ($\Delta$). We quantified $R_n$ and U
using monthly climatologies from the North American Land Data Assimilation System-2
(NALDAS-2) that were based on measurements from 1980-2009 (Mitchell et al., 2004). We
derived G, T, $e_s$, and $e_a$ from PRISM temperature data following Zotarelli et al. (2010). After
computing $CMI_{\overline{wy}}$, we then resampled these data using the nearest neighbor approach to match
the footprints of both the 1-km NPP and 250-m BIO remote sensing data sets.

**2.2    Analysis**
We quantified the response of forest NPP, BIO, and CRT to changes in $CMI_{\overline{wy}}$ across both
WAORCA and the broader western US. We specifically focused on areas where $CMI_{\overline{wy}}$ was
between -200 and 200 cm yr$^{-1}$, conditions which occurred both in WAORCA and in the broader
region. This range encompassed 98% of forest area in the western US; the paucity of data in the
remaining 2% of forest area that was either drier or wetter precluded rigorous analysis. We
divided the landscape along this gradient into 10 cm yr$^{-1}$ non-overlapping bins and then
summarized forest characteristics in each bin by computing the median, along with the 10$^{th}$, 25$^{th}$,
75$^{th}$ and 90$^{th}$ percentiles. Forest characteristics were summarized separately for the field and
remote sensing data sets. There were a minimum of 10 and a maximum of 114 field sites in each
bin. We then assessed the association between the median forest characteristic (i.e., NPP, BIO,
and CRT) in each bin and $CMI_{\overline{wy}}$ across the ecoclimatic gradient using nonparametric
Spearman's rank correlation. This test yields a coefficient ($r_s$) between -1 and +1, where a value





of +1 indicates a perfect monotonically increasing relationship, a value of zero indicates no
covariation between the two variables, and a value of -1 indicates a perfect monotonically
decreasing relationship. The test is analogous to Pearson's correlation where the data have first
been ranked. We assessed the association between forest characteristics and $CMI_{\overline{wy}}$ using
Spearman's correlation rather than nonlinear regression because our intent was to describe the
general relationship rather than develop a predictive model. We performed data preprocessing,
analysis, and visualization using ArcGIS 10 (ESRI, Redlands, CA) and *R* statistical software (R
Core Team, 2015), relying extensive on the *R* packages *raster* (Hijmans and van Etten, 2013)
and *dplyr* (Wickham and Francois, 2015).

### 2.3    Uncertainty

We minimize uncertainty in our analysis by incorporating satellite and field data sets, as well as
by examining the sensitivity of $CMI_{\overline{wy}}$ to methods used to estimate $ET_0$. Specifically, we
characterized forest BIO using three satellite-derived maps and field inventories. We similarly
characterized forest NPP using both satellite and field inventory data sets. This approach
combines the strengths of spatially continuous satellite-based model output with the rigor of
spatially-limited, field-based inventory measurements. Additionally, we computed $CMI_{\overline{wy}}$ based
on $ET_0$ derived using both the FAO-56 (Allen et al., 1998) and modified-Hargreaves (Hargreaves
and Samani, 1985;Droogers and Allen, 2002) methods. This comparison revealed that our results
were robust to differences in methods (results not shown) and thus we focused on $CMI_{\overline{wy}}$
computed using the FAO-56 method.



**3      Results**
Average annual climatic water availability varied widely across both WAORCA and the broader
western US from 1985-2014 (Fig. 1a, b). The $CMI_{\overline{wy}}$ ranged from around -400 cm $yr^{-1}$ in
southern California and Arizona to over 400 cm $yr^{-1}$ in the coastal mountain ranges in
northwestern Washington and Oregon. Forests mapped by MODIS occurred in areas where
$CMI_{\overline{wy}}$ was between -340 and 490 cm $yr^{-1}$, though 98% of forest area occurred between -200 and
200 cm $y^{-1}$, and 72% occurred between -100 and 100 cm $yr^{-1}$. Average (±1 SD) $CMI_{\overline{wy}}$ in
forested areas was -40±80 cm $yr^{-1}$. The Coast Range and Cascade Mountains in Washington and
Oregon were the wettest areas, with $CMI_{\overline{wy}}$ generally >100 cm $yr^{-1}$. Water availability decreased
rapidly in the rain shadows east of the Cascades and Sierra Nevada, giving rise to very steep
$CMI_{\overline{wy}}$ gradients. For instance, annual $CMI_{\overline{wy}}$ in northern Oregon decreased nearly 350 cm over
~30 km between high-elevation forests in the Cascades and low-elevation woodlands in the
eastern foothills of the Cascades. The range in $CMI_{\overline{wy}}$ encountered along this gradient in the
Cascades almost spanned the full range in $CMI_{\overline{wy}}$ that supported 98% of forest area in the
western US. Dry forests occurred along the low-elevation margins of mountain ranges
throughout continental areas, though the largest tract of dry forest was found in Arizona and New
Mexico.
Forest NPP, BIO, and BIO residence time varied substantially across both WAORCA and
the broader western US in response to variation in $CMI_{\overline{wy}}$ (Fig. 1, 2, Table 2). We focused on
forests in areas where $CMI_{\overline{wy}}$ was between -200 and 200 cm $yr^{-1}$ given the paucity of land and
measurements in the 2% of forest area that was either drier or wetter. Median $NPP_{field}$, $BIO_{field}$,
and $CRT_{field}$ all exhibited a strong, positive association with $CMI_{\overline{wy}}$ ($r_s$=0.93-0.96, p<0.001).





Median $NPP_{field}$ increased 155% between the driest and wettest 5% of sites in WAORCA (Fig.
2a), while median $BIO_{field}$ and $CRT_{field}$ increased 997% and 358%, respectively, between these
sites (Fig. 2b, c; Table 2). The relationship in each case was slightly curvilinear. There were also
strong, positive relationships among median $NPP_{field}$, $BIO_{field}$, and $CRT_{field}$ along the WAORCA
ecoclimatic gradient ($r_s$=0.90-0.96, p<0.001).

Broadly similar patterns were evident when forest $NPP_{sat}$, $BIO_{sat}$, and $CRT_{sat}$ were

examined across the western US using remote sensing data sets (Fig. 1b, c, d, 2c, d; Table 2).
Median $NPP_{sat}$, $BIO_{sat}$, and $CRT_{sat}$ all showed a strong, positive relationship with $CMI_{\overline{wy}}$
($r_s$=0.93-0.99; p<0.001). Median $NPP_{sat}$ increased 97% between the driest and wettest 5% of
forested areas along the regional $CMI_{\overline{wy}}$ gradient (Fig. 2d, Table 2). Similarly, median $BIO_{sat}$ and
$CRT_{sat}$ increased 410% and 160%, respectively, between the driest and wettest areas (Fig. 2e, f,
Table 2). The response of median $NPP_{sat}$, $BIO_{sat}$, and $CRT_{sat}$ to increased $CMI_{\overline{wy}}$ was more
curvilinear than the responses observed in the field measurements, with the satellite data sets
plateauing in areas where annual $CMI_{\overline{wy}}$ was above ~100 cm. Furthermore, while magnitude of
$NPP_{sat}$ and $NPP_{field}$ response to $CMI_{\overline{wy}}$ were similar, the magnitude of $BIO_{sat}$ and $CRT_{sat}$
responses to increased $CMI_{\overline{wy}}$ were much more muted than the magnitude of response in $BIO_{field}$
and $CRT_{field}$. There were again strong relationships among median $NPP_{sat}$, $BIO_{sat}$, and $CRT_{sat}$
along the western US ecoclimatic gradient ($r_s$=0.93-0.97, p<0.001).

**4      Discussion and conclusions**
**4.1      Climate moisture index**





Climatic water availability exerted a strong influence on NPP, BIO, and CRT among mature
forests in the western US. We chose to quantify climatic water availability using an index that
accounted for both precipitation and energy-mediated $ET_0$, recognizing that both of these factors
contribute to the relative water stress experienced by plants within an ecosystem (Webb et al.,
1983). We acknowledge that this index has several short-comings. For instance, the index does
not account for spatial variation in soil water storage capacity, which can be crucial for
determining plant performance during drought (Peterman et al., 2013). This might explain some
of the variation in NPP and BIO among areas with similar $CMI_{\overline{wy}}$; however, quantifying soil
water storage capacity even at individual sites is challenging given uncertainty in soil structure
and plant rooting capacity (Running, 1994). The index also does not account for water added via
fog drip, which has been shown to supply 13-45% of the water transpired by redwood forests (*S.
sempervirens*) (Dawson, 1998) and sustain other forest ecosystems along the California coast
(Fischer et al., 2016;Johnstone and Dawson, 2010). This potentially explains why there were
areas with low $CMI_{\overline{wy}}$ along the central and northern coast of California that supported forests
with higher NPP and BIO than other forests with similar $CMI_{\overline{wy}}$. Furthermore, the index does not
account for spatial variation in runoff and thus likely overestimates water availability in the
wettest areas since the fraction of water lost as run-off increases with precipitation (Sanford and
Selnick, 2013). Despite its relative simplicity, prior studies showed that CMI was a useful index
for explaining interannual variability in fire activity in the southwest US (Williams et al., 2014),
as well as forest productivity in northern Siberia (Berner et al., 2013), southern Canada (Hogg et
al., 2002), and central Oregon (Berner and Law, 2015). Several studies also found that the index,
or its inverse (i.e. $ET_0$ - P), explained substantial spatial variability in mature forest gross
photosynthesis (Law et al., 2002), productivity and biomass across a range of ecosystems



(Berner and Law, 2015;Webb et al., 1983;Hogg et al., 2008). Our current study further
demonstrates that CMI is a useful, empirical index for assessing climatic constraints on forest
ecosystems at large spatial scales.

## 4.2    Tree net primary productivity

Median forest NPP in mature stands approximately doubled between the driest and wettest areas
in both WAORCA and the western US, though in both cases the rate at which NPP increased
with $CMI_{\overline{wy}}$ slowed in the wettest areas. Prior field studies conducted at a limited number of field
sites in the western US over the past four decades have similarly documented increased forest
NPP along spatial gradients of increasing water availability (Webb et al., 1983;Whittaker and
Niering, 1975;Gholz, 1982;Berner and Law, 2015). Our current study demonstrates a robust
relationship between mature forest NPP and climatic water availability using field measurements
from nearly 2,000 inventory plots along with satellite remote sensing data sets covering ~18
Mha. The NPP-$CMI_{\overline{wy}}$ relationship was similar when NPP was assessed using field
measurements from across WAORCA or using MODIS covering the western US, though
MODIS showed NPP leveling off in the wettest areas ($CMI_{\overline{wy}} \approx$ 100-200 cm yr$^{-1}$), whereas this
was less evident in the field measurements. A recent remote sensing analysis of California used
absorbed photosynthetically-active radiation (APAR) derived from MODIS as an index of gross
primary productivity and found that APAR increased asymptotically with increasing mean
annual precipitation across vegetation communities (Jin and Goulden, 2014). Forests—
occupying the wettest areas and having the highest APAR—exhibited the smallest increase in
APAR per unit increase in precipitation of any vegetation community whether assessed along a



spatial or a temporal gradient, suggesting that forests were less sensitive to changes in
precipitation than other vegetation communities (Jin and Goulden, 2014). The lack of asymptotic
response in our field measurements together with the asymptotic response of both MODIS NPP
and APAR suggests that climate impact assessments based on MODIS could underestimate the
sensitivity of NPP to changes in water availability in wet, densely forested area.
Mechanistically, the strong NPP-CMI$_{\overline{wy}}$ association reflects the coupling between carbon
and water cycling at leaf (Ball et al., 1987) to ecosystem scales (Law et al., 2002). Forest NPP
depends on regionally-specific relations with leaf area (Waring, 1983;Schroeder et al., 1982),
which largely determine the proportion of incoming solar radiation that is absorbed and thus
potentially available to fuel photosynthesis (Runyon et al., 1994). Leaf photosynthesis inevitably
leads to transpiration water loss (Ball et al., 1987) that must be balanced against water uptake
from the soil so as to prevent the formation of excessive tension on the internal water column
that could result in hydraulic failure (Ruehr et al., 2014;Williams et al., 1996). As soil water
availability increases, trees are able to support greater leaf area while maintaining water column
tensions within physiologically operable ranges, which consequently leads to more
photosynthate available to fuel NPP unless trees are limited by other resources (e.g., nitrogen).
The decreasing rate at which NPP increased with CMI$_{\overline{wy}}$ in the wettest areas is likely due to low
temperatures constraining productivity at high-elevations (Nakawatase and Peterson,
2006;Runyon et al., 1994) and heavy cloud-cover limiting solar radiation and thus
photosynthesis in coastal areas (Carroll et al., 2014;Zhao et al., 2010). Forest NPP is affected by
many biotic (e.g., age) and abiotic factors (e.g., nutrients), yet climatic water availability emerges
as a key environmental constraint in the western US.






### 4.3    Tree carbon stocks

Mature forest BIO increased notably with increasing $CMI_{\overline{wy}}$ across both WAORCA and the
broader western US, reflecting underlying shifts in NPP and, likely, tree mortality due to
disturbance. BIO is determined by the rates at which carbon is gained via NPP and lost due to
tissue senescence and mortality integrated over annual to centennial time scales (Olson, 1963).
Hence, the increase in NPP with increasing $CMI_{\overline{wy}}$ explains some of the concomitant increase in
BIO. We suspect that as conditions became wetter there was also a decline in the proportion of
BIO lost to annual mortality from natural disturbances. Several recent studies found that tree
mortality rates due to bark beetles and fires were very low in the wettest parts of the western US
(e.g., Coast Range and Cascades), while considerably higher in most drier areas (Berner et al., in
review;Hicke et al., 2013). Furthermore, the field and satellite data sets also incidentally revealed
there was an increase in the median age of stands over 100 years as conditions became wetter,
with median stand age ~140 years in the driest areas and 200-240 years in the wettest areas. The
general increase in mature forest BIO with increasing water availability is thus likely due to
higher rates of productivity and lower rates of mortality from natural disturbance.

The observed increase in mature forest BIO with increasing climatic water availability

was generally consistent with prior field studies from this region, yet our study demonstrates this
response over a much broader ecoclimatic gradient. For instance, early work by Whittaker and
Niering (1975) showed that mature forest BIO tended to increase with a moisture index inferred
from community composition along an elevational gradient in Arizona's Santa Catalina
Mountains. Subsequent studies focused on five LTER sites spread across the conterminous US



(Webb et al., 1983) and at sites in Oregon (Berner and Law, 2015;Gholz, 1982) similarly showed
a general increase in tree biomass with increasing water availability. Our study included sites
that ranged from dry woodlands with little BIO to temperate rainforests with BIO exceeded in
few other regions (e.g. max BIO ≈ 950 Mg C ha$^{-1}$). BIO in our study area has been reported to
reach over 2,000 Mg C ha$^{-1}$ in old-growth coastal redwood stands in northern California (Waring
and Franklin, 1979), which is thought to be exceeded only by the >3,000 Mg C ha$^{-1}$ attained by
old-growth *Eucalyptus regnans* stands in southern Australia (Keith et al., 2009). A global
synthesis suggested that average AGB among high-biomass stands in wet temperate forests
(~377 Mg C ha$^{-1}$) was over twice that of high-biomass stands in wet tropical forests (~179 Mg C
ha$^{-1}$) and nearly six times that of high-biomass stands in wet boreal forests (~64 Mg C ha$^{-1}$)
(Keith et al., 2009). The range in mature forest BIO included in our analysis of WAORCA thus
spanned much of the observed global range in BIO.

Both field and satellite measurements revealed that median BIO increased with $CMI_{\overline{wy}}$,

yet the satellite data set showed less of an increase than the field measurements. Median forest
$BIO_{field}$ increased nearly 1,000% between the dry woodlands and coastal temperate rainforests in
WAORCA, yet the increase in $BIO_{sat}$ with increasing $CMI_{\overline{wy}}$ was less pronounced (410%
increase) when assessed across the western US. Furthermore, median $BIO_{sat}$ plateaued around
175 Mg C ha$^{-1}$ in areas where $CMI_{\overline{wy}}$ was ~100-200 cm yr$^{-1}$. The response of BIO to increasing
$CMI_{\overline{wy}}$ was likely more muted when assessed using the satellite-derived maps than the field
measurements for several reasons. Areas with high BIO often occur as small patches set in a
matrix of stands with lower BIO (Spies et al., 1994) and thus the moderate-resolution satellite
imagery used in developing these maps records the spectral signature of this larger area rather
than just the patch with high BIO. In other words, the satellite imagery has a larger sampling



footprint relative to that of a field plot, which thus averages BIO over a larger area, reducing
peak values. Additionally, the maps are largely derived from optical, multi-spectral satellite
imagery that is not very sensitive to variation in BIO in high-biomass forests. Advances in
satellite remote sensing, such as NASA's new Global Ecosystem Dynamics Investigation Lidar
(GEDI) instrument, are anticipated to help overcome some of these challenges (Goetz and
Dubayah, 2011). Nevertheless, current BIO maps (e.g., Wilson et al., 2013;Kellndorfer et al.,
2012) proved a valuable tool for ecologic and natural resource assessments (Goetz et al.,
2014;Krankina et al., 2014;Berner et al., in review).

**4.4    Carbon residence time in tree biomass**
Median $CRT_{field}$ increased persistently with $CMI_{\overline{wy}}$ from ~11 years in the driest forests to over 49
years in the wettest forests, highlighting a fundamental change in ecosystem function along this
broad ecoclimatic gradient. A prior study focused on 11 LTERS spread across the conterminous
US found that CRT increased from ~2 years in a desert shrubland to ~73 years in 450-years old
Douglas-fir stand at the Andrews LTER in the Oregon Cascade Mountains (Webb et al., 1983).
For comparison, we looked at five old-growth Douglas-fir stands (336-555 years old) near the
Andrews LTER and found that CRT averaged 79±23 years (± 1SD) among these stands. An
increase in the CRT of aboveground tissues was also observed among plant communities along
an elevational moisture gradient in the Arizona Santa Catalina Mountains (Whittaker and
Niering, 1975) and across nine mature stands in a range of forest types in Oregon (Gholz, 1982).
Although this pattern has been documented in several instances, the underlying mechanisms
remain unclear.



We speculate that the increase in CRT with increased water availability was potentially
associated with underlying changes in NPP allocation and BIO mortality rates. Trees invest a
larger proportion of NPP into aboveground tissue production as conditions become wetter and
competition for light intensifies (Runyon et al., 1994;Law et al., 2003). Our field measurements
revealed that the fraction of NPP allocated aboveground increased from ~0.45 in the driest areas
to ~0.64 in the wettest areas and, furthermore, that CRT in aboveground tissues averaged twice
as long as the CRT in belowground tissues. Thus, a shift in NPP allocation toward longer-lived
aboveground tissues likely contributed to the observed increase in CRT as conditions become
wetter, as might changes in BIO mortality rates along this hydraulic gradient. Recent BIO
mortality rates due to disturbance by wildfires and bark beetles tended to be considerably lower
in the wettest parts of the western US than in drier parts of the region (Berner et al., in review).
The incidental observation that mature stands tended to be older in the wetter areas is consistent
with these areas experiencing lower morality rates from natural disturbances. Our study
demonstrates that CRT in live tree biomass was strongly influenced by water availability, yet
additional efforts are needed to elucidate underlying mechanism affecting CRT, particularly
given that CRT is a primary source of uncertainty in global vegetation model projections of
future terrestrial carbon cycling (Friend et al., 2014).

**4.5     Climate change implications**
Forest NPP, BIO, and CRT in mature stands increased with $CMI_{\overline{wy}}$ across WAORCA and the
broader western US, underscoring that climatic water availability is a major abiotic constraint on
several keys aspects of ecosystem structure and function in forests ranging from dry woodlands





to coastal temperate rainforests. What do these findings mean in the context of regional climate
change? Although future changes in precipitation are uncertain, climate models widely project
extensive regional warming over the coming century in response to high rates of greenhouse gas
emissions, which could lead to drier conditions as higher temperatures increase atmospheric
evaporation demand (Walsh et al., 2014;Collins et al., 2013;Dai, 2013). For instance, simulations
based on the sophisticated Variable Infiltration Capacity (VIC) hydraulic model and a high-
emission scenario (A2) suggest that soil moisture could decline ~1-15% in many parts of the
region by the end of the 21$^{st}$ century, with drying particularly acute in the Southwest (Walsh et
al., 2014). Similarly, a large ensemble of climate models indicate that soil moisture could decline
3-12% throughout the region over this century (Dai, 2013). In fact, projections of regional drying
are widespread, particularly for the Southwest (e.g., Williams et al., 2012;Schwalm et al.,
2012;Burke et al., 2006;Seager and Vecchi, 2010;Dai, 2011;Collins et al., 2013).

Increased atmospheric $CO_2$ and warming in the Northwest could enhance tree

productivity in some areas by (1) increasing water use efficiency (WUE) through $CO_2$
fertilization and (2) enhancing spring photosynthesis (Ruehr et al., 2014;Hudiburg et al.,
2013;Kang et al., 2014;Soulé and Knapp, 2015). On the other hand, many tree species have
narrow hydraulic safety margins (Choat et al., 2012) and warming-induced declines in tree
growth have occurred in other regions despite increased WUE (Andreu-Hayles et al.,
2011;Peñuelas et al., 2011;Lévesque et al., 2014). It is unlikely that increased WUE and other
physiological adjustments will fully compensate for impacts of rapid future warming on tree
physiology (Allen et al., 2015), especially in the Southwest where hotter and drier conditions are
already suppressing tree productivity and increasing tree mortality in some areas (Dennison et
al., 2014;Creeden et al., 2014;Williams et al., 2012;Anderegg et al., 2015;McDowell et al.,



2015). The strong NPP-CMI$_{\overline{wy}}$, BIO-CMI$_{\overline{wy}}$, and CRT-CMI$_{\overline{wy}}$ associations that we observed in
the western US suggest that future reductions in water availability will likely reduce NPP, BIO,
and CRT in mature forests, particularly those in the driest areas.

**4.6    Conclusions**
Forests in the western US range from dry woodlands to temperate rainforests, an ecological
gradient that nearly spans the global range in tree biomass and that largely reflects spatial
variation in climatic water availability. In this study, we quantified changes in tree productivity,
live biomass, and carbon residence time along spatial gradients in climatic water availability
using field inventory measurements from WAORCA and satellite remote sensing data sets
spanning the western US. Our multi-method, multi-scale analysis revealed that tree productivity,
live biomass, and carbon residence time all increased notably with climatic water availability,
which was computed using an index that accounted for both precipitation and reference
evapotranspiration. The observed increase in productivity was likely due to the close coupling
between carbon and water cycling at leaf to ecosystem scales, while the observed increase in live
biomass was likely due to the increased productivity and stand age, along with a decreased
proportion of live biomass lost to annual mortality. Forest productivity and biomass derived from
field- and satellite-measurements exhibited broadly similar sensitivities to changes in climatic
water availability, though the satellite data sets tended to plateau in the wettest areas, suggesting
that additional efforts are needed to better quantify productivity and biomass from satellites in
high-productivity, high-biomass forests. The pronounced increase in carbon residence time with
increasing water availability suggests that efforts to increase terrestrial carbon storage as a tool to




combat climate change will be most effective in the wettest areas. Furthermore, the observed
change in carbon residence time could provide a benchmark for evaluating the performance of
global vegetation models, in which carbon residence time is a principle source of uncertainty in
future projections of the global carbon cycle. Overall, our results indicate that tree productivity,
live biomass, and carbon residence time in mature stands are widely sensitive to changes in
climatic water availability in the western US, suggesting that projected warming and drying over
the coming century due to business-as-usual greenhouse gas emissions could have important
impacts on ecosystem structure, function, and services in many parts of this region.

**Author contributions**
L.T.B. designed the study, analyzed the data, and prepared the manuscript with contributions
from B.E.L. and T.W.H., who both also contributed data sets to this effort.

**Acknowledgements**
This work was supported by NASA Headquarters under the NASA Earth and Space Science
Fellowship Program (Grant NNX14AN65H), the USDA National Institute of Food and
Agriculture (Grant 2013-67003-20652), and the ARCS Foundation Scholar program. T.W.H.
was supported by the National Science Foundation Idaho EPSCoR Program (Grant IIA-
1301792). We thank the researchers who provided the geospatial data sets used in this analysis.
We cite no conflicts of interest.





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



**Tables**
**Table 1.** Summary of tree net primary productivity (NPP; Mg C ha$^{-1}$ yr$^{-1}$), live biomass (BIO; Mg C ha$^{-1}$),
and carbon residence time (CRT; year) for stands over 100 years old across both WAORCA and the
broader western US. These forest characteristics were quantified for WAORCA using field measurements
from 1,953 sites and for the western US using satellite-derived data sets covering 18 Mha of mature
forest. Satellite data sets included MODIS NPP and an estimate of BIO derived by combining existing
maps of aboveground biomass with additional estimates of carbon storage in coarse root, fine roots, and
foliage. CRT describes the average duration, in years, that a molecule of carbon will remain in live tree
biomass and was computed as CRT=BIO/NPP. It is also known as the biomass accumulation ratio. These
carbon stocks and fluxes combine above- and below-ground components.

| Domain | Variable | Units | Time span | Mean (SD) | Range |
|---|---|---|---|---|---|
| WAORCA | NPP$_{field}$ | Mg C ha$^{-1}$ yr$^{-1}$ | 2001-2006 | 4.3 (2.5) | 0.6 – 20.9 |
| | BIO$_{field}$ | Mg C ha$^{-1}$ | 2001-2006 | 158 (135) | 2 – 947 |
| | CRT$_{field}$ | year | 2001-2006 | 33 (19) | 2 – 137 |
| Western US | NPP$_{sat}$ | Mg C ha$^{-1}$ yr$^{-1}$ | 2000-2014 | 5.3 (2.0) | 0.1 – 227 |
| | BIO$_{sat}$ | Mg C ha$^{-1}$ | 2000-2008 | 83 (54) | 2 – 669 |
| | CRT$_{sat}$ | year | 2000-2008 | 15 (9) | 2 – 1390 |














**Table 2.** Changes in tree net primary productivity (NPP; Mg C ha$^{-1}$ yr$^{-1}$), live biomass (BIO; Mg
C ha$^{-1}$), and carbon residence time (CRT; year) for stands over 100 years old along gradients in a
climate moisture index (CMI$_{\overline{wy}}$; cm yr$^{-1}$) in both WAORCA and the broader western US. Forest
characteristics were quantified using field measurements in WAORCA and satellite remote
sensing data sets covering the western US. The analysis incorporated forests in areas where
CMI$_{\overline{wy}}$ was between -200 cm yr$^{-1}$ and 200 cm yr$^{-1}$. Summaries include (1) median forest
characteristic in the driest 5% and wettest 95% of sites/pixels; (2) the corresponding change; (3)
and the Spearman correlation ($r_s$) between CMI$_{\overline{wy}}$ and the median forest characteristic computed
at 10 cm yr$^{-1}$ CMI$_{\overline{wy}}$ intervals. All correlations were statistically significant at $\alpha < 0.001$.

| Domain | Variable | Units | Median of… | | Change… | | CMI$_{\overline{wy}}$ cor. |
|--------|----------|-------|------------|------------|------|-----|------------|
| | | | Driest 5% | Wettest 95% | Abs. | % | $r_s$ |
| WAORCA | NPP$_{field}$ | Mg C ha$^{-1}$ yr$^{-1}$ | 2.2 | 5.6 | 3.4 | 155 | 0.93 |
| | BIO$_{field}$ | Mg C ha$^{-1}$ | 26 | 281 | 255 | 997 | 0.96 |
| | CRT$_{field}$ | year | 11 | 49 | 38 | 358 | 0.96 |
| Western US | NPP$_{sat}$ | Mg C ha$^{-1}$ yr$^{-1}$ | 3.4 | 6.7 | 3.3 | 97 | 0.93 |
| | BIO$_{sat}$ | Mg C ha$^{-1}$ | 32 | 165 | 133 | 410 | 0.97 |
| | CRT$_{sat}$ | year | 10 | 26 | 16 | 160 | 0.99 |















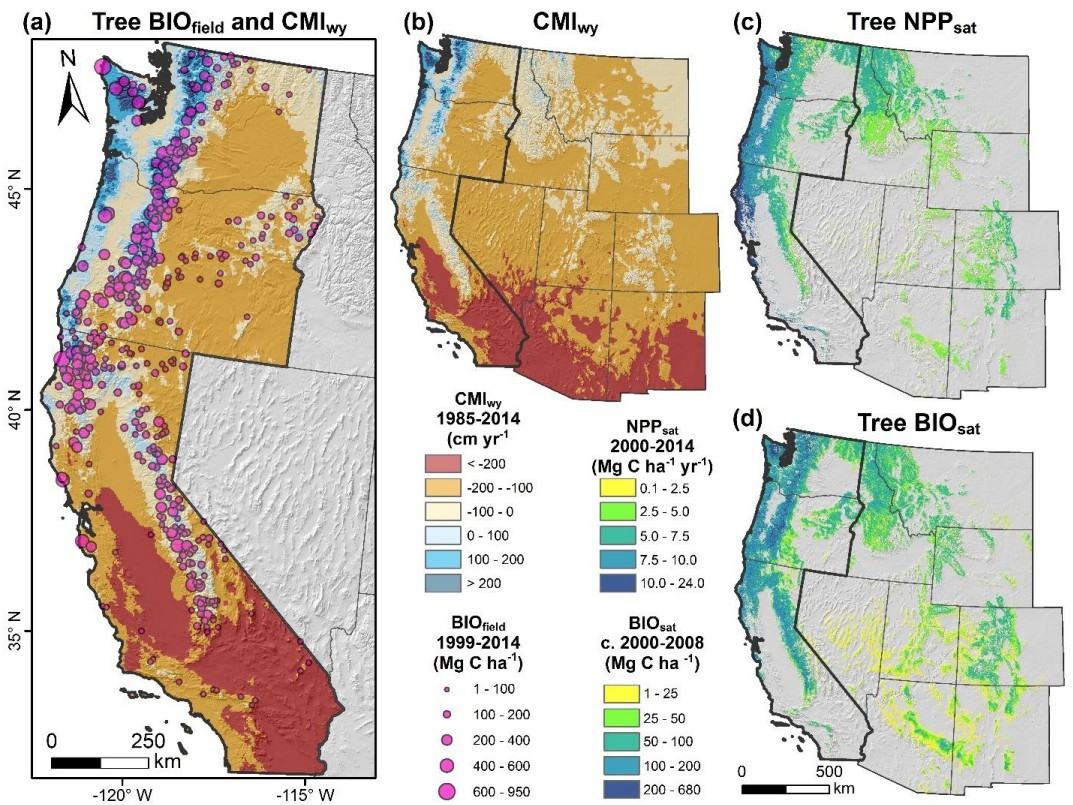


**Figure 1.** Mean climatic moisture index ($\mathrm{CMI}_{\overline{wy}}$; cm yr$^{-1}$), tree net primary productivity (NPP;

Mg C ha$^{-1}$ yr$^{-1}$), and live tree biomass (BIO; Mg C ha$^{-1}$) in the western US. (a) BIO derived from

field measurements (BIO$_{field}$) at mature sites (>100 years) in WAORCA. For visual clarity only

20% of the 1,953 sites are depicted. (b) $\mathrm{CMI}_{\overline{wy}}$ was computed as monthly precipitation minus

reference evapotranspiration summed over the annual water year (October-September) and then

averaged from 1985-2014. (c) Mean annual NPP was quantified using MODIS satellite data from

2000-2014 (NPP$_{sat}$). (d) BIO was quantified using satellite-derived estimates of carbon stocks

(BIO$_{sat}$).




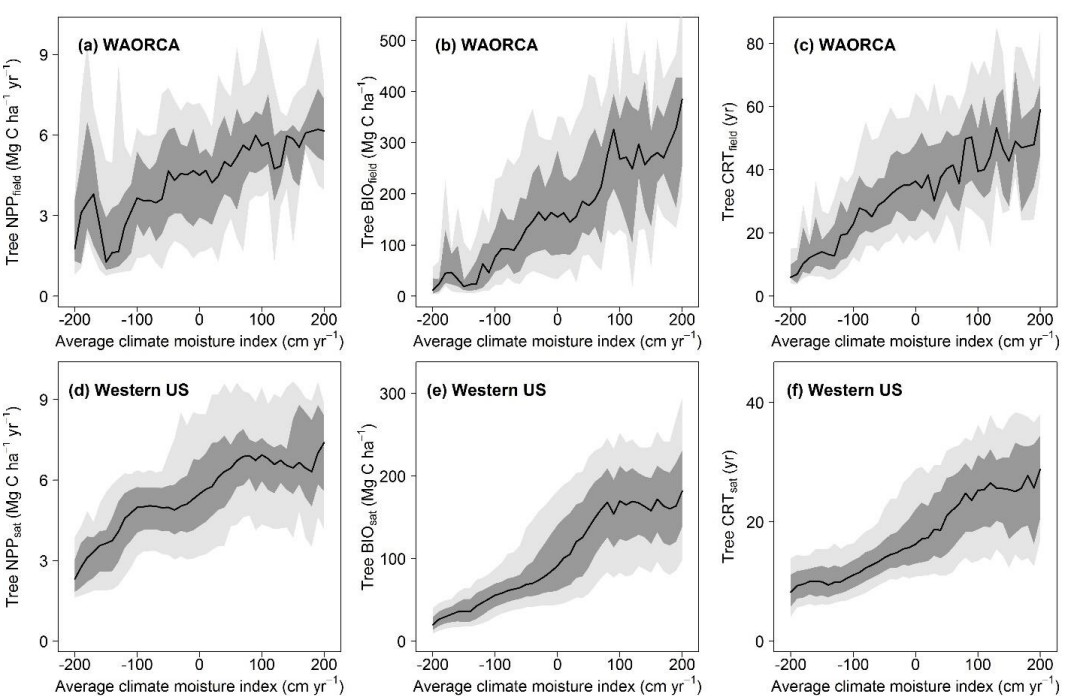


**Figure 2.** Tree net primary productivity (NPP; Mg C ha$^{-1}$ yr$^{-1}$), live biomass (BIO; Mg C ha$^{-1}$),

and carbon residence time (CRT; years) increased with increasing climatic water availability
across both WAORCA (a-c) and the broader western US (d-f). Forest characteristics were
derived from field measurements on 1,953 inventory plots in WAORCA (a-c) and from satellite
remote sensing data sets across 18 Mha of mature forest in the western US (d-f). NPP$_{sat}$ was
characterized using MODIS data averaged annual from 2000 to 2014. BIO$_{sat}$ was quantified
based on an ensemble of aboveground biomass maps plus estimates of coarse root, fine root, and
foliage biomass. CRT was computed for each field plot and pixel as BIO / NPP. Water
availability was quantified using a climate moisture index (CMI= P-ET$_0$) summed over the water
year (October-September) and then averaged from 1985-2014 (CMI$_{\overline{wy}}$). The region was
partitioned into 10 cm yr$^{-1}$ (non-overlapping) CMI$_{\overline{wy}}$ bins, pixels/plots were allocated to bins,
and then forest characteristics were summarized within each bin. In each panel, the bold line





denotes the median, dark gray band the 25-75$^{th}$ percentiles, and light gray band the 10-90$^{th}$
percentiles. Note the different y-axis scales between (b) and (e), as well as (c) and (f).