# Peer review of "Water availability limits tree productivity, carbon stocks, and carbon residence time in mature forests across the western United States"

_Biogeosciences, 2016_

## Referee Comment (RC1) · Anonymous Referee #1 · 7 Nov 2016

General Comments:

This is a very interesting paper that uses forest inventory and satellite data to evaluate the influence of mean annual moisture balance on forest productivity and biomass across the western US with a particular focus on California, Oregon, and Washington. While it is no surprise that productivity and biomass in this region are affected by water availability, this paper provides the most thorough quantification of this influence to date and represents a fantastic use of US Forest Service survey data. This thorough quantification leads to the conclusion that water balance has not just an important influence on forest carbon in the western US, but that it is instead THE dominant driver in this region, with a strong and reliable effect on both biomass and productivity, which

translates to a strong and reliable effect on carbon residence time. This information has clear implications for future forest carbon dynamics in a warmer world with altered precipitation regimes, which is importance since dynamic vegetation models are still in need of substantial improvement before their representations of future shifts in forest demographics across regions as large and complex as the western US are taken seriously. I recommend publication after some minor points below are addressed.

Specific Comments:

L79: The Singh et al. study is a great one but the focus is not on the impact of recent warm temperatures on west coast drought, but rather on an observed increase in the frequency of east-west dipole years when the western US is anomalously warm and the eastern US is anomalously cool. There have been many papers that more compellingly evaluate the role of temperature in exacerbating recent drought conditions on the west coast, particularly CA, than either of the references provided here:

AghaKouchak, A., L. Cheng, O. Mazdiyasni, A. Farahmand (2014), Global warming and changes in risk of concurrent climate extremes: Insights from the 2014 California drought, Geophysical Research Letters, 41(24), 8847-8852, doi:10.1002/2014GL062308.

Griffin, D., K. J. Anchukaitis (2014), How unusual is the 2012–2014 California drought? Geophysical Research Letters, 41(24), 9017-9023, doi:10.1002/2014GL062433.

Mao, Y., B. Nijssen, D. P. Lettenmaier (2015), Is climate change implicated in the 2013-2014 California drought? A hydrologic perspective, Geophysical Research Letters, 42(8), 2805-2813, doi:10.1002/2015GL063456.

Mote, P. W., D. E. Rupp, S. Li, D. J. Sharp, F. Otto, P. F. Uhe, M. Xiao, D. P. Lettenmaier, H. Cullen, M. R. Allen (2016), Perspectives on the causes of exceptionally low 2015 snowpack in the western United States, Geophysical Research Letters, 10.1002/2016GL069965, In press, doi:10.1002/2016GL069965.

[Figure]

Shukla, S., M. Safeeq, A. AghaKouchak, K. Guan, C. Funk (2015), Temperature impacts on the water year 2014 drought in California, Geophysical Research Letters, 42(11), 4384-4393, doi:10.1002/2015GL063666.

Williams, A. P., R. Seager, J. T. Abatzoglou, B. I. Cook, J. E. Smerdon, E. R. Cook (2015), Contribution of anthropogenic warming to California drought during 2012-2014, Geophysical Research Letters, 42(16), 6819-6828, doi:10.1002/2015GL064924.

L132-136: The alometric equations and LAI-vs-root relationship should be cited, particularly for the diverse (non-forestry) readership of this journal.

L140: I don't think it's necessary to specify that stands of >100 years of age are considered here since it was already stated that only stands of this age group were considered in the analysis.

L177-194: The circularity involved in using the MODIS NPP product, which incorporates climate data, to evaluate the relationship between NPP and climate needs to be acknowledged.

L341-344: Is this artifact due to saturation of satellite-derived NDVI/LAI in densely vegetated areas? It seems like the likely reason for the false plateauing in the satellite obs could be stated.

L459-460: The projected soil moisture trends in Dai (2013) are for just 0-10 cm. For model projections of the more important 1-2 m layer, Cook et al. (2015) is a good reference, at least for CA and the Southwest.

Cook, B. I., T. R. Ault, J. E. Smerdon (2015), Unprecedented 21st century drought risk in the American Southwest and Central Plains, Science Advances, 1(1), e1400082, doi:10.1126/sciadv.1400082.

L469-472: But isn't it under hotter/drier conditions where, all else held equal, vegetation stands to benefit the most from increased CO2. The argument that recent drought-driven declines in productivity in the Southwest is evidence for a lack of a CO2

effect is an incomplete argument, as it could be counter-argued that the recent drought period has been particularly intense and that the consequences would have been more severe without CO2 fertilization. There is still much that is unknown about CO2 fertilization, the forests that will benefit from it, and how these benefits will manifest, but just as it is unwise to argue that CO2 fertilization will definitely allow semi-arid forests to become more productive in a warmer world, it is also unwise to imply without a thorough evaluation of evidence that CO2 fertilization will not have any effect on the future relationship between CMI and NPP, BIO, or CRT.

Technical Corrections:

L39 & 41: CMI should be defined on L39, therefore allowing the definition of CMIwy on L41 to make more sense.

L94: Should "be" be "by"?

L112: The specification of the converse hypothesis is unnecessary.

L125: Should "using" be "used"?

L237: Should "extensive" be "extensively"?

L443: "elucidate underlying mechanism" may be missing a word or letter.

Fig 2 caption, L847: Should "annual" be "annually"?

---

## Referee Comment (RC2) · Anonymous Referee #2 · 21 Nov 2016

This is a nice study demonstrating the regional relationship between water availability and productivity, C stocks and residence time in forests of the western US. An impressive data set based on both forest inventory and satellite data were used to establish these relationships. I am not a specialist in estimating forest NPP or C dynamics, but the methods used and assumptions made seem reasonable and the authors are experts in these ecosystems.

Their results indicate that mature forests in the western US were strongly sensitive (across spatial gradients) to changes in water availability. This is not a surprising result, but the scale and scope of this analysis makes this a publishable study. Where I take issue is the inference drawn from this analysis. The authors conclude that their

analysis suggests that projected climatic change over the coming century could reduce productivity, biomass and carbon residence time in many parts of this region. Indeed, they justify their study by noting that "Changes in ecosystem structure and function along spatial climatic gradients can provide insight into long-term ecosystem response to climatic change". While this makes sense in the broadest terms, using spatial relationships (based on average values derived from long-term data) to make predictions about temporal changes in (or the differential sensitivity of) ecosystems to a climate change is risky at best.

We have long known that large scale spatial relationships between NPP and precipitation (or water availability) have a slope that is determined by combined changes in water availability, biogeochemistry and the plant community. But the temporal dynamics over which each of these factors will change in the future will vary dramatically. . .from decades to centuries to even millennia. Thus, spatial models of NPP vs. water are not good predictors of expected temporal dynamics in ecosystems. . ..particularly in forests that have long-lived trees and where communities may turnover very slowly (hundreds of years?). Please see the three references below. Combined, they do a nice job of covering many of the well-known problems inherent in substituting spatial models for temporal models when projecting a future with directional and chronic climate change.

Thus, while I am in favor of publishing this analysis, the conclusions drawn that "projected warming and drying over the coming century. . .could have important impacts on ecosystem structure, function, and services. . ." are really not that noteworthy. Nonetheless, a well-done confirmatory message is much better than much of the introduction and discussion which repeatedly references "sensitivities to changes in water availability" in the context of climate change. As presented, the implication that there is climate change relevance in this analysis is really quite misleading. . .given that spatial sensitivity does not equate to temporal sensitivity – except perhaps for sign. This is true under today's environment, and spatial relationships such as those derived here will likely be even poorer surrogates for predicting the future as the varying time scales of change

(climate vs forest community turnover vs. biogeochemistry) lead to novel functional relationships.

Thus, at the very least the authors should point out the limitations of their analysis and approach with regard to its relevance to future temporal C dynamics. Specifically, because the slopes (sensitivity) of temporal relationships between NPP and water are almost always less steep than slopes from spatial models, the authors need to recognize that the sensitivity implied by their analysis will likely not be manifest.

Estiarte et al. 2016. Few multi‐year precipitation‐reduction experiments find a shift in the productivity‐precipitation relationship. Global Change Biology 22: 2570–2581.

Gaitan et al. 2014. Vegetation structure is as important as climate for explaining ecosystem function across Patagonian rangelands. Journal of Ecology 102: 1419-1428.

Wilcox 2016. Does ecosystem sensitivity to precipitation at the site-level conform to regional-scale predictions? Ecology 97: 561-568.

---

## Referee Comment (RC3) · Anonymous Referee #3 · 21 Nov 2016

In my view, this paper makes an important contribution in quantifying the relationship between forest characteristics (net primary productivity, NPP; live biomass, BIO; mean carbon residence time, CRT) and climatic moisture regimes in the western United States. The analysis is strengthened by the inclusion of two fundamentally different data sources and methods, including forest inventory measurements from 3 states (WA, OR and CA) and satellite-based estimates across an even larger area (11 western states). The results are striking as both methods show that forest characteristics in this region are governed primarily by spatial gradients in climatic moisture regimes (as represented by a simple climate moisture index, CMI_wy). Although this general conclusion is not new, the work provides valuable quantitative estimates of forest-climate

relationships that are likely to be useful in improving models of forest responses to the climatic drying that is already evident in this region. Overall, the paper is clearly presented and the methods seem appropriate, given the major challenges of spatial scaling in this mountainous and climatically diverse region.

My main questions relate to a) the justification for including only mature stands > 100 years, and b) unstated assumptions and potential sources of error in estimating CRT (see specific points, below).

Specific comments:

L110 What is the justification for restricting the analysis to mature stands older than 100 years? Is this age considered to be a threshold, beyond which the variables BIO, NPP and/or CRT remain constant over time?

L127-128: Do these inventory sites represent forests across the full range of elevations in this region? If they exclude sampling of unproductive forests in climatically cold, wet sites near the upper timberline then I'm wondering if this could explain the observed differences in response to CMI for inventory sites versus satellite-derived estimates (Fig. 2).

L138-140: I believe that the equation used for carbon residence time (CRT = BIO / NPP) is based on the assumption that BIO is constant over time (e.g., see equation 1 of Friend et al 2014, reference cited in the MS). If so, then this assumption (and any others) should be stated explicitly. Overall, I'm wondering how much of the variation in the reported estimates of CRT is driven simply by variation in stand age, given that in my experience, older stands tend to exhibit increasing (or stable) values of BIO along with age-related declines in NPP. On a related point, using the above equation, I would expect estimates of CRT to be inflated in forests with anomalously low NPP over the 10-year period of calculation (e.g., during droughts or insect defoliation episodes that are not sufficiently severe to cause a proportional decline in BIO). Again, it would be helpful to at least acknowledge the potential sources of bias in the reported CRT

estimates.

L396-400: The discussion includes reporting of the large percentage difference in BIO across the climatic moisture gradient (CMI_wy) using the two methods (from Fig. 2) but I expect that the percentage difference would be even greater than this if dry, naturally unforested areas (with zero forest BIO and NPP) were included in the analysis. In this respect, it would be interesting to see how %forest cover varies as a function of the binned values of CMI_wy across this region. I recognize that such an analysis would go beyond the scope of this paper, but it could provide an interesting additional indicator of how forest NPP and carbon stocks may respond over the long term under the projected (and ongoing) climatic drying, i.e., drought-related loss of forest cover in addition to drought-related decreases in BIO, NPP and CRT in those sites that continue to remain forested.

―――――――――――――――――――――

---

## Referee Comment (RC4) · Anonymous Referee #4 · 29 Nov 2016

General comments: This is a thorough, straightforward study using both field and satellite measurements to estimate forest productivity and carbon cycling along a spatial moisture index across the western US. The goals of the study were outlined well, and made use of two datasets that if assimilated properly, can reveal ecological trends and relationships that cross spatial scales. The results revealed, unsurprisingly, that as moisture index increased, so did both productivity and biomass; however this study is one of the more thorough I have seen in both its spatial and methodological scale. The results suggest that climatic moisture availability is perhaps the most fundamental environmental control of forests in the Western US, and that the forest communities are extremely sensitive to this across large spatial scales.

[Figure]

I feel this study is well conceived and publishable, but needs more explanation of methods, particularly with regards to data assimilation and validation. You mention in section 2.3 that you 'minimize[d] uncertainty' by using two different data types (field and remotely sensed), but you present no evidence of this. Also, though you present the Spearman coefficient in Table 2, I would have liked to see some cross-domain validation between data types; that is, a simple statistical comparison of how each median variable (NPP, BIO, CRT) value compares between field and satellite data.

Specific comments: L52: Mention of ecosystem services seems unnecessary

L69: Suggest substituting 'risk' with 'frequency' or 'occurrence'

L101: CRT should be defined before acronym is introduced.

L154: This sentence is very unclear. I don't understand what 'ensemble average' is referring to, nor what the 'previous work' revealed.

L196: Should it be climate 'data' sets?

L196: Some context should be given for CMI values. What is the typical range? What constitutes extreme values on either end?

L229: Make sure use of 'Spearman' or 'Spearman's' is consistent

L447: Changing natural disturbance regimes should be mentioned in the climate change implications section, given that you discuss it earlier in the context of carbon residence time.

---

## Editor Comment (EC1) · C. Bourque (Editor) · 6 Dec 2016

I would like to take opportunity to thank you for your input; you provide many useful suggestions. The authors should take into account of your comments in their revision of the manuscript.

Best regards,

Charles P.-A. Bourque

---

## Editor Comment (EC2) · C. Bourque (Editor) · 6 Dec 2016

I would like to take opportunity to thank you for your input. I agree with you that the limitations of the study need to be made more explicit. I trust the authors will take this into account in their revision of the manuscript.

Best regards,

Charles P.-A. Bourque

---

## Editor Comment (EC3) · C. Bourque (Editor) · 6 Dec 2016

Thank you for your input. The authors should strive to implement your suggestions in their revision of the manuscript.

Best regards,

Charles P.-A. Bourque

---

## Editor Comment (EC4) · C. Bourque (Editor) · 6 Dec 2016

Thank you for your input. I appreciate your participation with the review of the manuscript. The authors should consider your comments in their revision of the manuscript.

---

## Author Response (AR1)

**Author responses to referee comments**

**Manuscript ID**: bg-2016-419

**Manuscript title**: Water availability limits tree productivity, carbon stocks, and carbon residence time in mature forests across the western United States

**Authors:** Logan T. Berner, Beverly E. Law, and Tara W. Hudiburg

**Anonymous Referee #1**

**General Comments**:

This is a very interesting paper that uses forest inventory and satellite data to evaluate the influence of mean annual moisture balance on forest productivity and biomass across the western US with a particular focus on California, Oregon, and Washington. While it is no surprise that productivity and biomass in this region are affected by water availability, this paper provides the most thorough quantification of this influence to date and represents a fantastic use of US Forest Service survey data. This thorough quantification leads to the conclusion that water balance has not just an important influence on forest carbon in the western US, but that it is instead THE dominant driver in this region, with a strong and reliable effect on both biomass and productivity, which translates to a strong and reliable effect on carbon residence time. This information has clear implications for future forest carbon dynamics in a warmer world with altered precipitation regimes, which is importance since dynamic vegetation models are still in need of substantial improvement before their representations of future shifts in forest demographics across regions as large and complex as the western US are taken seriously. I recommend publication after some minor points below are addressed.

RESPONSE: Thank you for your feedback on our manuscript. Your comments were very helpful in preparing a revised version of our manuscript. We made the minor revisions that you suggested, as detailed below.

**Specific Comments**:

L79: The Singh et al. study is a great one but the focus is not on the impact of recent warm temperatures on west coast drought, but rather on an observed increase in the frequency of east-west dipole years when the western US is anomalously warm and the eastern US is anomalously cool. There have been many papers that more compellingly evaluate the role of temperature in exacerbating recent drought conditions on the west coast, particularly CA, than either of the references provided here:

AghaKouchak, A., L. Cheng, O. Mazdiyasni, A. Farahmand (2014), Global warming and changes in risk of concurrent climate extremes: Insights from the 2014 California drought, Geophysical Research Letters, 41(24), 8847-8852, doi:10.1002/2014GL062308.

Griffin, D., K. J. Anchukaitis (2014), How unusual is the 2012–2014 California drought? Geophysical Research Letters, 41(24), 9017-9023, doi:10.1002/2014GL062433.

Mao, Y., B. Nijssen, D. P. Lettenmaier (2015), Is climate change implicated in the 2013- 2014 California drought? A hydrologic perspective, Geophysical Research Letters, 42(8), 2805-2813, doi:10.1002/2015GL063456.

Mote, P. W., D. E. Rupp, S. Li, D. J. Sharp, F. Otto, P. F. Uhe, M. Xiao, D. P. Lettenmaier, H. Cullen, M. R. Allen (2016), Perspectives on the causes of exceptionally low 2015 snowpack in the western United States, Geophysical Research Letters, 10.1002/2016GL069965, In press, doi:10.1002/2016GL069965.

Shukla, S., M. Safeeq, A. AghaKouchak, K. Guan, C. Funk (2015), Temperature impacts on the water year 2014 drought in California, Geophysical Research Letters, 42(11), 4384-4393, doi:10.1002/2015GL063666.

Williams, A. P., R. Seager, J. T. Abatzoglou, B. I. Cook, J. E. Smerdon, E. R. Cook (2015), Contribution of anthropogenic warming to California drought during 2012-2014, Geophysical Research Letters, 42(16), 6819-6828, doi:10.1002/2015GL064924.

RESPONSE: Thank you for guiding us towards several more appropriate references. We removed the citation to Singh et al. (2016) found several of the suggested citations to be more appropriate (e.g., Mote et al. 2016; AghaKouchak et al. 2014).

L132-136: The allometric equations and LAI-vs-root relationship should be cited, particularly for the diverse (non-forestry) readership of this journal.

RESPONSE: We added the appropriate citations for the allometric equations (Means et al. 1994; Law et al. 2001) and equation for estimating root biomass from leaf area index (Van Tuyl et al. 2005).

L140: I don’t think it’s necessary to specify that stands of >100 years of age are considered here since it was already stated that only stands of this age group were considered in the analysis.

RESPONSE: Suggestion adopted.

L177-194: The circularity involved in using the MODIS NPP product, which incorporates climate data, to evaluate the relationship between NPP and climate needs to be acknowledged.

RESPONSE: We agree that the circularity should be acknowledged and added the following sentence to the methods (section 2.1.2):

> We acknowledge a degree of circularity in relating $NPP_{sat}$ to CMI given that both computations incorporate temperature data, specifically, temperature-effects on VPD.

L341-344: Is this artifact due to saturation of satellite-derived NDVI/LAI in densely vegetated areas? It seems like the likely reason for the false plateauing in the satellite obs could be stated.

RESPONSE: It is possible that the apparent saturation of MODIS NPP in the wettest areas was related to MODIS becoming less sensitive to variation in FPAR in densely vegetated areas. We modified part of the text to read (starting on line 337):

> The NPP-$CMI_{\overline{wy}}$ relationship was similar when NPP was assessed using field measurements from across WAORCA or using MODIS covering the western US. MODIS did show NPP leveling off in the wettest parts of WAORCA ($CMI_{\overline{wy}} \approx$ 100-200 cm $yr^{-1}$), whereas this was less evident in the field measurements. The inventory sites and MODIS forestland occurred at similar elevations along the $CMI_{\overline{wy}}$ gradient in WAORCA, suggesting that this discrepancy in NPP was not due to MODIS systematically including cold, high-elevation areas not sampled by the inventory sites. One possibility is that MODIS NPP did not increase in the wettest areas because MODIS becomes less sensitive to increases in the fraction of photosynthetically-active radiation (FPAR) absorbed by plant canopies in densely vegetated areas (Yan et al., 2016)….

L459-460: The projected soil moisture trends in Dai (2013) are for just 0-10 cm. For model projections of the more important 1-2 m layer, Cook et al. (2015) is a good reference, at least for CA and the Southwest. Cook, B. I., T. R. Ault, J. E. Smerdon (2015), Unprecedented 21st century drought risk in the American Southwest and Central Plains, Science Advances, 1(1), e1400082, doi:10.1126/sciadv.1400082.

RESPONSE: Thank you for the suggestion. We incorporated this reference into our discussion.

L469-472: But isn't it under hotter/drier conditions where, all else held equal, vegetation stands to benefit the most from increased CO2. The argument that recent drought-driven declines in productivity in the Southwest is evidence for a lack of a CO2 effect is an incomplete argument, as it could be counter-argued that the recent drought period has been particularly intense and that the consequences would have been more severe without CO2 fertilization. There is still much that is unknown about CO2 fertilization, the forests that will benefit from it, and how these benefits will manifest, but just as it is unwise to argue that CO2 fertilization will definitely allow semi-arid forests to become more productive in a warmer world, it is also unwise to imply

without a thorough evaluation of evidence that CO2 fertilization will not have any effect on the future relationship between CMI and NPP, BIO, or CRT.

RESPONSE: Following this comment and feedback from Reviewer 2, we chose to remove the 'Climate change implications' section and replace it with a section called 'Predicting ecosystem response to environmental change' that reads (starting on line 475):

> Water availability is projected to decline in much of the western US over the coming century, in part due to higher temperatures increasing atmospheric evaporative demand (Walsh et al., 2014;Dai, 2013;Cook et al., 2015). Predicting the timing, magnitude and extent of ecological response to regional climate change remains a challenge. Our study showed that water availability is a key determinant of forest structure and function in the western US, broadly suggesting that chronic reductions in regional water availability could reduce the NPP, BIO, and CRT of mature stands. Nevertheless, it is problematic to predict the temporal response of extant forest communities to near-term climatic change based on ecoclimatic relationships derived from spatial data. For instance, recent studies found that the slope of the NPP-precipitation relationship was much steeper when derived from spatial data than when derived from the temporal response of NPP to interannual variation in precipitation (Wilcox et al., 2016;Jin and Goulden, 2014). Near-term effects of climate variability depend on the physiological characteristics of species in the extant plant community, yet ecoclimatic relationships derived from spatial data reflect gradual adjustment of community composition and population size to climate over long periods of time (Wilcox et al., 2016;Jin and Goulden, 2014). Furthermore, ecoclimatic models derived from spatial data cannot account for other ecophysiological impacts of environmental change, such as (1) enhanced plant water use efficiency from $CO_2$ fertilization (Soulé and Knapp, 2015); (2) increased likelihood of tree mortality due to hotter drought (Adams et al., 2009); or (3) novel changes in disturbance regimes (Hicke et al., 2006;Dale et al., 2001). Consequently, predicting ecological response to environmental change over the coming century will require the use of mechanistic ecosystem models that account for physiologic, demographic, and disturbance processes at fine taxonomic and spatial scales (Law, 2014;Hudiburg et al., 2013). Although spatial models may not be suitable for near-term projection of ecosystems change, they do provide insight into long-term ecosystem adaptation to local climate and, furthermore, can be used to validate and refine mechanistic models if constructed from a representative sample of forestlands.

**Technical Corrections**:

L39 & 41: CMI should be defined on L39, therefore allowing the definition of CMIwy on L41 to make more sense.

RESPONSE: Suggestion adopted.

L94: Should “be” be “by”?

RESPONSE: Yes, thank you.

L112: The specification of the converse hypothesis is unnecessary.

RESPONSE: The converse hypothesis has been removed.

L125: Should “using” be “used”?

RESPONSE: Yes, thank you.

L237: Should “extensive” be “extensively”?

RESPONSE: Yes, thank you.

L443: “elucidate underlying mechanism” may be missing a word or letter.

RESPONSE: We changed the sentence to read, “…additional efforts are needed to determine the underlying mechanism by which changes in water availability affect CRT.”

L847 (Fig 2 caption): Should “annual” be “annually”?

RESPONSE: Yes, thank you.

**Anonymous Referee #2**

This is a nice study demonstrating the regional relationship between water availability and productivity, C stocks and residence time in forests of the western US. An impressive data set based on both forest inventory and satellite data were used to establish these relationships. I am not a specialist in estimating forest NPP or C dynamics, but the methods used and assumptions made seem reasonable and the authors are experts in these ecosystems.

Their results indicate that mature forests in the western US were strongly sensitive (across spatial gradients) to changes in water availability. This is not a surprising result, but the scale and scope of this analysis makes this a publishable study. Where I take issue is the inference drawn from this analysis. The authors conclude that their analysis suggests that projected climatic change

over the coming century could reduce productivity, biomass and carbon residence time in many parts of this region. Indeed, they justify their study by noting that "Changes in ecosystem structure and function along spatial climatic gradients can provide insight into long-term ecosystem response to climatic change". While this makes sense in the broadest terms, using spatial relationships (based on average values derived from long-term data) to make predictions about temporal changes in (or the differential sensitivity of) ecosystems to a climate change is risky at best.

We have long known that large scale spatial relationships between NPP and precipitation (or water availability) have a slope that is determined by combined changes in water availability, biogeochemistry and the plant community. But the temporal dynamics over which each of these factors will change in the future will vary dramatically. . .from decades to centuries to even millennia. Thus, spatial models of NPP vs. water are not good predictors of expected temporal dynamics in ecosystems...particularly in forests that have long-lived trees and where communities may turnover very slowly (hundreds of years?). Please see the three references below. Combined, they do a nice job of covering many of the well-known problems inherent in substituting spatial models for temporal models when projecting a future with directional and chronic climate change.

Thus, while I am in favor of publishing this analysis, the conclusions drawn that "projected warming and drying over the coming century. . .could have important impacts on ecosystem structure, function, and services. . ." are really not that noteworthy. Nonetheless, a well-done confirmatory message is much better than much of the introduction and discussion which repeatedly references "sensitivities to changes in water availability" in the context of climate change. As presented, the implication that there is climate change relevance in this analysis is really quite misleading. . .given that spatial sensitivity does not equate to temporal sensitivity – except perhaps for sign. This is true under today's environment, and spatial relationships such as those derived here will likely be even poorer surrogates for predicting the future as the varying time scales of change (climate vs forest community turnover vs. biogeochemistry) lead to novel functional relationships.

Thus, at the very least the authors should point out the limitations of their analysis and approach with regard to its relevance to future temporal C dynamics. Specifically, because the slopes (sensitivity) of temporal relationships between NPP and water are almost always less steep than slopes from spatial models, the authors need to recognize that the sensitivity implied by their analysis will likely not be manifest.

Estiarte, M., Vicca, S., Peñuelas, J., Bahn, M., Beier, C., Emmett, B. A., Fay, P. A., Hanson, P. J., Hasibeder, R., and Kigel, J.: Few multi□year precipitation□reduction experiments find a shift in the productivity□precipitation relationship, Global change biology, 2016.

Gaitan et al. 2014. Vegetation structure is as important as climate for explaining ecosystem function across Patagonian rangelands. Journal of Ecology 102: 1419- 1428.

Wilcox 2016. Does ecosystem sensitivity to precipitation at the site-level conform to regional-scale predictions? Ecology 97: 561-568.

RESPONSE: We appreciate your critique of our manuscript, as well as the references that you suggested. We revised our manuscript to better acknowledge that the ecoclimatic relationships we observed reflect long-term climatic constraints on ecosystem structure and function, which are shaped by gradual shifts in community composition and population size (Jin and Goulden, 2014). Consequently, these ecoclimatic relationships are not sufficient to predict ecosystem response to near-term changes in climate. We re-wrote the introduction, de-emphasizing observed and projected climate change, while emphasizing how this study seeks to confirm earlier observation at a larger scale. Furthermore, we remove the "Climate change implications" section (4.5) from the discussion and replaced it with section called "Predicting ecosystems response to environmental change" that reads (starting on line 475):

"Water availability is projected to decline in much of the western US over the coming century, in part due to higher temperatures increasing atmospheric evaporative demand (Walsh et al., 2014;Dai, 2013;Cook et al., 2015). Predicting the timing, magnitude and extent of ecological response to regional climate change remains a challenge. Our study showed that water availability is a key determinant of forest structure and function in the western US, broadly suggesting that chronic reductions in regional water availability could reduce the NPP, BIO, and CRT of mature stands. Nevertheless, it is problematic to predict the temporal response of extant forest communities to near-term climatic change based on ecoclimatic relationships derived from spatial data. For instance, recent studies found that the slope of the NPP-precipitation relationship was much steeper when derived from spatial data than when derived from the temporal response of NPP to interannual variation in precipitation (Wilcox et al., 2016;Jin and Goulden, 2014). Near-term effects of climate variability depend on the physiological characteristics of species in the extant plant community, yet ecoclimatic relationships derived from spatial data reflect gradual adjustment of community composition and population size to climate over long periods of time (Wilcox et al., 2016;Jin and Goulden, 2014). Furthermore, ecoclimatic models derived from spatial data cannot account for other ecophysiological impacts of environmental change, such as (1) enhanced plant water use efficiency from $CO_2$ fertilization (Soulé and Knapp, 2015); (2) increased likelihood of tree mortality due to hotter drought (Adams

et al., 2009); or (3) novel changes in disturbance regimes (Hicke et al., 2006;Dale et al., 2001). Consequently, predicting ecological response to environmental change over the coming century will require the use of mechanistic ecosystem models that account for physiologic, demographic, and disturbance processes at fine taxonomic and spatial scales (Law, 2014;Hudiburg et al., 2013). Although spatial models may not be suitable for near-term projection of ecosystems change, they do provide insight into long-term ecosystem adaptation to local climate and, furthermore, can be used to validate and refine mechanistic models if constructed from a representative sample of forestlands."

We also modified the Summary and Conclusions section to read (starting on line 519):

"The pronounced increase in tree productivity, biomass, and carbon residence time between the driest and wettest areas illustrates the gradual adjustment of ecosystem structure and function to long-term variation in water availability; however, the observed ecoclimatic relationships are not suitable for near-term projections of future ecosystem response to regional drying. Predicting near-term ecosystem response to drying and other environmental change (e.g., increased $CO_2$) will require mechanistic ecosystem models, which can be evaluated against ecoclimatic relationships developed using inventory sites from a representative sample of forestlands (e.g., Forest Service inventory sites). Overall, our results indicate long-term water availability is a key determinant of tree productivity, live biomass, and carbon residence time in mature stands ranging from dry woodlands to coastal temperate rainforests, underscoring that additional efforts are needed to anticipate and mitigate the impacts of projected warming and drying on forest ecosystems in the western US and elsewhere around the world."

**Anonymous Referee #3**

**General comments:**

In my view, this paper makes an important contribution in quantifying the relationship between forest characteristics (net primary productivity, NPP; live biomass, BIO; mean carbon residence time, CRT) and climatic moisture regimes in the western United States. The analysis is strengthened by the inclusion of two fundamentally different data sources and methods, including forest inventory measurements from 3 states (WA, OR and CA) and satellite-based estimates across an even larger area (11 western states). The results are striking as both methods show that forest characteristics in this region are governed primarily by spatial gradients in climatic moisture regimes (as represented by a simple climate moisture index, CMI_wy). Although this general conclusion is not new, the work provides valuable quantitative estimates of forest-climate relationships that are likely to be useful in improving models of forest responses to

the climatic drying that is already evident in this region. Overall, the paper is clearly presented and the methods seem appropriate, given the major challenges of spatial scaling in this mountainous and climatically diverse region.

My main questions relate to a) the justification for including only mature stands > 100 years, and b) unstated assumptions and potential sources of error in estimating CRT (see specific points, below).

RESPONSE: Thank you for providing valuable comments on our manuscript. In the revised manuscript we provide (1) better justification for focusing on mature stands and (2) a more thorough discussion of the assumptions and limitations associated with computing CRT as BIO/NPP. We describe these revisions in greater detail below.

**Specific comments:**

L110 What is the justification for restricting the analysis to mature stands older than 100 years? Is this age considered to be a threshold, beyond which the variables BIO, NPP and/or CRT remain constant over time?

RESPONSE: We focused on mature stands (>100 years) because inventory plots in this region showed that tree BIO and NPP tended to increase rapidly with stand age during the first century and then change more gradually during subsequent years (Hudiburg et al., 2009). In essence, we assume that BIO and NPP have hit much of their 'climatic potential' after 100 years. Several prior studies similarly focused on mature stands that were at least 100 years old (Gholz, 1982;Whittaker and Niering, 1975;Webb et al., 1983). Furthermore, computing CRT as BIO/NPP assumes (as discussed below) that BIO is stable through time, which is an assumption met more closely by examining older stands. We did perform the analysis using all forestland, regardless of stand age, and found very similar results (albeit with lower BIO and CRT). We added the following text to the introduction (starting on line 76):

"Prior studies drew on small networks of field sites (n < 20) to investigate how tree net primary productivity (NPP) and BIO varied among mature stands spread along hydrologic gradients in parts of this region (Webb et al., 1983;Berner and Law, 2015;Whittaker and Niering, 1975;Gholz, 1982). Tree BIO and NPP can vary widely with stand age (Hudiburg et al., 2009)

and thus these studies focused on mature stands (stand age generally > 100 years) where BIO and NPP had somewhat stabilized after reaching their 'climatic potential.'"

We also added the following text to the introduction directly before stating our hypotheses (starting on line 112):

"We focused on forest stands that were at least 100 years old because field surveys from the region indicated that BIO and NPP reached much of their 'climatic potential' after a century, though we acknowledge that BIO tends to gradually increase and NPP remains stable or gradually declines during subsequent centuries (Hudiburg et al., 2009)."

L127-128: Do these inventory sites represent forests across the full range of elevations in this region? If they exclude sampling of unproductive forests in climatically cold, wet sites near the upper timberline then I'm wondering if this could explain the observed differences in response to CMI for inventory sites versus satellite-derived estimates (Fig. 2).

RESPONSE: An astute question. The Forest Service inventory sites are spread among areas > 1 acre (0.40 ha) that have at least 10% tree cover (Bechtold and Patterson, 2005). The sites do occur in cold, wet, high-elevation areas so long as those requirements are met. We compared the average (SD) elevation of inventory sites and MODIS forest pixels (stands > 100 years) at each step along the CMIwy gradient and found that inventory sites and MODIS forest had very similar elevational distribution across WAORCA. A paired t-test found no significant difference in average elevation between inventory sites and MODIS forest along the CMIwy gradient in WAORCA (P=0.43). The differences between inventory and satellite-derived estimates of BIO and NPP were most apparent in the wettest areas (e.g, CMIwy > 100 cm/yr) that overwhelming occurred in WAORCA (e.g., 98% of MODIS forest with CMIwy > 100 cm/yr was in WAORCA). Consequently, it does not appear that differences between inventory and satellite-derived estimates of BIO and NPP in wet areas can be attributed to the satellite data systematically including cold, high-elevation areas that are were not represented by the inventory sites. We added text in several places to help clarify. We edited part of the methods to read (starting on line 134):

"These 1-ha sites were surveyed by the US Forest Service Forest Inventory and Analysis (FIA) program between 2001 to 2006 and comprise a representative sample of forest lands (tree cover > 10%) in the region (Bechtold and Patterson, 2005). The inventory sites occurred at elevations ranging from 5 m to 3,504 m, with an average (±1SD) elevation of 1429±677 m."

Additionally, we added modified part of the discussion to read (starting on line 336):

"The NPP-$\mathrm{CMI}_{\overline{\mathrm{wy}}}$ relationship was similar when NPP was assessed using field measurements from across WAORCA or using MODIS covering the western US, though MODIS did show NPP leveling off in the wettest parts of WAORCA ($\mathrm{CMI}_{\overline{\mathrm{wy}}} \approx$ 100-200 cm $yr^{-1}$), whereas this was less evident in the field measurements. The inventory sites and MODIS forestland occurred at similar elevations along the $\mathrm{CMI}_{\overline{\mathrm{wy}}}$ gradient in WAORCA, suggesting that this discrepancy in NPP was not due to MODIS systematically including cold, high-elevation areas not sampled by the inventory sites. One possibility is that MODIS NPP did not increase in the wettest areas because MODIS becomes less sensitive to increases in the fraction of photosynthetically-active radiation (FPAR) absorbed by plant canopies in densely vegetated areas (Yan et al., 2016)."

L138-140: I believe that the equation used for carbon residence time (CRT = BIO / NPP) is based on the assumption that BIO is constant over time (e.g., see equation 1 of Friend et al 2014, reference cited in the MS). If so, then this assumption (and any others) should be stated explicitly. Overall, I'm wondering how much of the variation in the reported estimates of CRT is driven simply by variation in stand age, given that in my experience, older stands tend to exhibit increasing (or stable) values of BIO along with age-related declines in NPP. On a related point, using the above equation, I would expect estimates of CRT to be inflated in forests with anomalously low NPP over the 10-year period of calculation (e.g., during droughts or insect defoliation episodes that are not sufficiently severe to cause a proportional decline in BIO). Again, it would be helpful to at least acknowledge the potential sources of bias in the reported CRT estimates.

RESPONSE: You are correct that calculating mean carbon residence time (CRT) as CRT = BIO/NPP assumes that BIO is constant over time and we agree that this assumption should be explicitly stated and more thoroughly discussed in our manuscript. We added the following text to the introduction to note this assumption (starting on line 91):

"Several of these earlier field studies also indicated that plant communities accumulated more BIO per unit of NPP in progressively wetter areas, suggesting slower turnover of plant BIO as climate became wetter (Webb et al., 1983;Whittaker and Niering, 1975). Mean carbon residence time (CRT) describes the average duration that a carbon molecule will remain in a specific pool (Waring and Running, 2007) and for CRT in live biomass can be computed as BIO/NPP assuming that BIO remains constant over time (Friend et al., 2014;Whittaker, 1961). CRT in live biomass is also known as the *biomass accumulation ratio* (Whittaker, 1961) and ranged, for instance, from ~2 years in a hot desert shrubland to ~75 years in an wet, old-growth Douglas-fir forest (Webb et al., 1983)."

Older stands did have higher CRT and average stand age did increase moving into wetter areas. Together, these indicate that the CRT-CMIwy relationships we observed did incorporate an age-related effect; however, the age-relate effect appears to be rather small. For instance, let's compare the median CRT between mature (100-200 years) and old (>200 years) stands occupying very dry (CMIwy < -100 cm $yr^{-1}$) and very wet (CMIwy > 100 cm $yr^{-1}$) areas. Median CRT differed by 6% (16 vs. 17 years) between mature and old stands in very dry areas and by 10% (47 vs. 52 years) in very wet areas. Conversely, median CRT of mature stands differed 98% (16 vs. 47 years) between very dry and very wet areas, while the median CRT of old stands differed 101% (52 vs. 17 years) between very dry and very wet areas. In very dry areas 80% of stands were mature and 20% were old, whereas in very wet areas 67% of stands were mature and 33% were old. Furthermore, CRT-CMIwy relationships constructed using mature and old were quite similar, diverging slightly in the wettest areas. These comparisons illustrate CRT is affected by stand age, but that the age effect is quite small relative to the climate effect. We edited the CRT discussion section so that it now begins by addressing uncertainty in our estimates of CRT (starting on line 430):

"One limitation of our study is that computing CRT in this manner assumes that BIO is constant over time (Friend et al., 2014). We focused on mature stands (>100 years) to minimize the change in BIO over time, though acknowledge that BIO can gradually increase during subsequent centuries (Hudiburg et al., 2009), which would lead us to underestimated CRT. Conversely, drought and insect-induced defoliation in the early 2000s could have suppressed NPP (Berner and Law, 2015;Schwalm et al., 2012) without a proportional reduction in BIO, which could have inflated our estimates of CRT in some areas."

We then revised the text to include a discussion of the age-related effect (starting on line 457):

"We also found that mature stands tended to be older in wetter areas and that older stands tended to have longer CRT, likely as a result of these stands having higher BIO and similar NPP (Hudiburg et al., 2009). Consequently, the CRT-$CMI_{\overline{wy}}$ relationships that we observed incorporate an age-related effect; however, the effect was quite small relative to the climate-effect. This can be illustrated by comparing median CRT between mature (100-200 years) and old (>200 years) stands occupying very dry ($CMI_{\overline{wy}}$< -100 cm $yr^{-1}$) and very wet ($CMI_{\overline{wy}}$ > 100 cm $yr^{-1}$) areas. Median CRT differed by 6% (16 vs. 17 years) between mature and old stands in very dry areas and by 10% (47 vs. 52 years) in very wet areas. Conversely, median CRT of mature stands differed 98% (16 vs. 47 years) between very dry and very wet areas, while the median CRT of old stands differed 101% (52 vs. 17 years) between very dry and very wet areas. In other words, the difference in CRT between stands in contrasting climates is much greater than difference in CRT between mature and old stands within a climate zone. Our study

demonstrates that CRT in live tree biomass was strongly influenced by water availability, yet additional efforts are needed to determine the underlying mechanism by which changes in water availability affect CRT, particularly given that CRT is a primary source of uncertainty in global vegetation model projections of future terrestrial carbon cycling (Friend et al., 2014)."

L396-400: The discussion includes reporting of the large percentage difference in BIO across the climatic moisture gradient (CMI_wy) using the two methods (from Fig. 2) but I expect that the percentage difference would be even greater than this if dry, naturally unforested areas (with zero forest BIO and NPP) were included in the analysis. In this respect, it would be interesting to see how %forest cover varies as a function of the binned values of CMI_wy across this region. I recognize that such an analysis would go beyond the scope of this paper, but it could provide an interesting additional indicator of how forest NPP and carbon stocks may respond over the long term under the projected (and ongoing) climatic drying, i.e., drought-related loss of forest cover in addition to drought-related decreases in BIO, NPP and CRT in those sites that continue to remain forested.

RESPONSE: We appreciate the suggestion and believe that it would be interesting to investigate how forest cover changes with CMIwy over this region; however, we believe this addition is beyond the scope of our current study.

**Anonymous Referee #4**

**General comments:**

This is a thorough, straightforward study using both field and satellite measurements to estimate forest productivity and carbon cycling along a spatial moisture index across the western US. The goals of the study were outlined well, and made use of two datasets that if assimilated properly, can reveal ecological trends and relationships that cross spatial scales. The results revealed, unsurprisingly, that as moisture index increased, so did both productivity and biomass; however this study is one of the more thorough I have seen in both its spatial and methodological scale. The results suggest that climatic moisture availability is perhaps the most fundamental environmental control of forests in the Western US, and that the forest communities are extremely sensitive to this across large spatial scales. I feel this study is well conceived and publishable, but needs more explanation of methods, particularly with regards to data assimilation and validation. You mention in section 2.3 that you 'minimize[d] uncertainty' by using two different data types (field and remotely sensed), but you present no evidence of this. Also, though you present the Spearman coefficient in Table 2, I would have liked to see some

cross-domain validation between data types; that is, a simple statistical comparison of how each median variable (NPP, BIO, CRT) value compares between field and satellite data.

RESPONSE: We appreciate the reviewer taking the time to comment on our manuscript. In the revised manuscript we removed the comment about 'minimizing uncertainty by incorporating both field and remote sensing data sets,' which was not phrased appropriately. In fact, we removed that section (2.3) entirely and incorporated select element into other parts of the manuscript. Following the reviewer's second comment, we compared field- and satellite-derived estimates of median NPP, BIO, and CRT, which showed that they were strongly correlated. We then added a sentence towards the end of the results section stating that, "Field- and satellite-derived estimates of median NPP, BIO, and CRT were strongly correlated ($r_s$=0.90-0.95; $p<0.001$)." We address the reviewer's remaining comments below.

**Specific comments:**

L52: Mention of ecosystem services seems unnecessary

RESPONSE: We removed the reference to ecosystem services.

L69: Suggest substituting 'risk' with 'frequency' or 'occurrence'

RESPONSE: We changes 'risk' to 'occurrence.'

L101: CRT should be defined before acronym is introduced.

RESPONSE: We edited these sentences to read (starting on line 91):

"Several of these earlier field studies also indicated that plant communities accumulated more BIO per unit of NPP in progressively wetter areas, suggesting slower turnover of plant BIO as climate became wetter (Webb et al., 1983;Whittaker and Niering, 1975). Mean carbon residence time (CRT) describes the average duration that a carbon molecule will remain in a specific pool (Waring and Running, 2007) and for CRT in live biomass can be computed as BIO/NPP assuming that BIO remains constant over time (Friend et al., 2014;Whittaker, 1961)."

L154: This sentence is very unclear. I don't understand what 'ensemble average' is referring to, nor what the 'previous work' revealed.

RESPONSE: We changed the two sentences to read (starting on line 161):

"We then reprojected these maps onto a uniform grid in an equal area projection, masked them to the common forest extent, and then averaged the AGB for each pixel across the three biomass maps. We used the biomass map ensemble average in the subsequent analysis, recognizing that pixel-wise estimates of AGC can vary notably among individual maps (Neeti and Kennedy, 2016)."

L196: Should it be climate 'data' sets?

RESPONSE: Yes, thank you.

L196: Some context should be given for CMI values. What is the typical range? What constitutes extreme values on either end?

RESPONSE: We provide a summary of minimum and maximum CMIwy across the western US, as well as the average CMIwy in forested areas in the results section.

L229: Make sure use of 'Spearman' or 'Spearman's' is consistent

RESPONSE: We edited the manuscript to consistently use *Spearman's* .

L447: Changing natural disturbance regimes should be mentioned in the climate change implications section, given that you discuss it earlier in the context of carbon residence time.

RESPONSE: We ended up replacing the 'Climate change implication' section with a section called 'Predicting ecosystems response to environmental change,' which mentions the importance of changes in disturbance regimes. Part of this section reads (starting on line 485):

"Near-term effects of climate variability depend on the physiological characteristics of species in the extant plant community, yet ecoclimatic relationships derived from spatial data reflect

gradual adjustment of community composition and population size to climate over long periods of time (Wilcox et al., 2016;Jin and Goulden, 2014). Furthermore, ecoclimatic models derived from spatial data cannot account for other ecophysiological impacts of environmental change, such as (1) enhanced plant water use efficiency from $CO_2$ fertilization (Soulé and Knapp, 2015); (2) increased likelihood of tree mortality due to hotter drought (Adams et al., 2009); or (3) novel changes in disturbance regimes (Hicke et al., 2006;Dale et al., 2001). Consequently, predicting ecological response to environmental change over the coming century will require the use of mechanistic ecosystem models that account for physiologic, demographic, and disturbance processes at fine taxonomic and spatial scales (Law, 2014;Hudiburg et al., 2013)."

**Citations**

Adams, H. D., Guardiola-Claramonte, M., Barron-Gafford, G. A., Villegas, J. C., Breshears, D. D., Zou, C. B., Troch, P. A., and Huxman, T. E.: Temperature sensitivity of drought-induced tree mortality portends increased regional die-off under global-change-type drought, Proceedings of the National Academy of Sciences of the United States of America, 106, 7063-7066, 10.1073/pnas.0901438106, 2009.

Bechtold, W., and Patterson, P.: The enhanced forest inventory and analysis program-- national sampling design and estimation procedures, USDA Forest Service General Technical Report SRS-80, Asheville, NC, 98, 2005.

Berner, L. T., and Law, B. E.: Water limitations on forest carbon cycling and conifer traits along a steep climatic gradient in the Cascade Mountains, Oregon, Biogeosciences, 12, 6617-6635, 2015.

Cook, B. I., Ault, T. R., and Smerdon, J. E.: Unprecedented 21st century drought risk in the American Southwest and Central Plains, Science Advances, 1, e1400082, 2015.

Dai, A.: Increasing drought under global warming in observations and models, Nature Climate Change, 3, 52-58, doi: 10.1038/nclimate1633, 2013.

Dale, V. H., Joyce, L. A., McNulty, S., Neilson, R. P., Ayres, M. P., Flannigan, M. D., Hanson, P. J., Irland, L. C., Lugo, A. E., Peterson, C. J., Simberloff, D., Swanson, F. J., Stocks, B. J., and Michael Wotton, B.: Climate Change and Forest Disturbances, BioScience, 51, 723-734, 10.1641/0006-3568(2001)051[0723:ccafd]2.0.co;2, 2001.

Friend, A. D., Lucht, W., Rademacher, T. T., Keribin, R., Betts, R., Cadule, P., Ciais, P., Clark, D. B., Dankers, R., and Falloon, P. D.: Carbon residence time dominates uncertainty in terrestrial vegetation responses to future climate and atmospheric CO2, Proceedings of the National Academy of Sciences, 111, 3280-3285, 2014.

Gholz, H. L.: Environmental Limits on Aboveground Net Primary Production, Leaf Area, and Biomass in Vegetation Zones of the Pacific Northwest, Ecology, 63, 469-481, 10.2307/1938964, 1982.

Hicke, J. A., Logan, J. A., Powell, J., and Ojima, D. S.: Changing temperatures influence suitability for modeled mountain pine beetle (Dendroctonus ponderosae) outbreaks in the western United States, Journal of Geophysical Research: Biogeosciences (2005–2012), 111, 2006.

Hudiburg, T., Law, B., Turner, D. P., Campbell, J., Donato, D., and Duane, M.: Carbon dynamics of Oregon and Northern California forests and potential land-based carbon storage, Ecological Applications, 19, 163-180, 2009.

Hudiburg, T. W., Law, B. E., and Thornton, P. E.: Evaluation and improvement of the Community Land Model (CLM4) in Oregon forests, Biogeosciences, 10, 453-470, 10.5194/bg-10-453-2013, 2013.

Jin, Y., and Goulden, M. L.: Ecological consequences of variation in precipitation: separating short-versus long-term effects using satellite data, Global Ecology and Biogeography, 23, 358-370, 2014.

Law, B. E.: Regional analysis of drought and heat impacts on forests: current and future science directions, Global Change Biology, 20, 3595-3599, 10.1111/gcb.12651, 2014.
Neeti, N., and Kennedy, R.: Comparison of national level biomass maps for conterminous US: understanding pattern and causes of differences, Carbon Balance and Management, 11, 19, 10.1186/s13021-016-0060-y, 2016.
Schwalm, C. R., Williams, C. A., Schaefer, K., Baldocchi, D., Black, T. A., Goldstein, A. H., Law, B. E., Oechel, W. C., and Scott, R. L.: Reduction in carbon uptake during turn of the century drought in western North America, Nature Geoscience, 5, 551-556, 2012.
Soulé, P. T., and Knapp, P. A.: Analyses of intrinsic water-use efficiency indicate performance differences of ponderosa pine and Douglas-fir in response to CO2 enrichment, Journal of Biogeography, 42, 144-155, 2015.
Walsh, J., Wuebbles, D., Hayhoe, K., Kossin, J., Kunkel, K., Stephens, G., Thorne, P., Vose, R., Wehner, M., Willis, J., Anderson, D., Doney, S., Feely, R., Hennon, P., Kharin, V., Knutson, T., Landerer, F., Lenton, T., Kennedy, J., and Somerville, R.: Our Changing Climate, in: Climate Change Impacts in the United States: The Third National Climate Assessment, edited by: Melillo, J. M., Richmond, T. C., and Yohe, G. W., U.S. Global Change Research Program, 19-67, 2014.
Waring, R. H., and Running, S. W.: Forest Ecosystems: Analysis at Multiple Scales, 3rd ed., Elsevier Academic Press, Burlington, MA, 420 pp., 2007.
Webb, W. L., Lauenroth, W. K., Szarek, S. R., and Kinerson, R. S.: Primary production and abiotic controls in forests, grasslands, and desert ecosystems in the United States, Ecology, 134-151, 1983.
Whittaker, R. H.: Estimation of net primary production of forest and shrub communities, Ecology, 42, 177-180, 1961.
Whittaker, R. H., and Niering, W. A.: Vegetation of the Santa Catalina Mountains, Arizona. V. Biomass, production, and diversity along the elevation gradient, Ecology, 56, 771-790, 1975.
Wilcox, K. R., Blair, J. M., Smith, M. D., and Knapp, A. K.: Does ecosystem sensitivity to precipitation at the site-level conform to regional-scale predictions?, Ecology, 97, 561-568, 2016.
Yan, K., Park, T., Yan, G., Liu, Z., Yang, B., Chen, C., Nemani, R. R., Knyazikhin, Y., and Myneni, R. B.: Evaluation of MODIS LAI/FPAR Product Collection 6. Part 2: Validation and Intercomparison, Remote Sensing, 8, 460, 2016.

**List of relevant changes**

We made the following changes to our manuscript during the revision process, as described in greater detail in the section above. This list only includes substantive changes, with changes listed in the same order as our responses to the reviewer's comments:

1. We added an acknowledgement regarding the circularity of comparing MODIS NPP against the CMI given that both include the effect of VPD.
2. We added text discussing possible reasons why MODIS NPP could saturate in the wettest areas.
3. We replaced the 'Climate change implications' section in the discussion with a section called 'Predicting ecosystem response to environmental change."
4. We revised our manuscript to better acknowledge that the ecoclimatic relationships we observed reflect long-term climatic constraints on ecosystem structure and function, which are shaped by gradual shifts in community composition and population size (Jin and Goulden, 2014). Consequently, these ecoclimatic relationships are not sufficient to predict ecosystem response

to near-term changes in climate. We re-wrote the introduction, de-emphasizing observed and projected climate change, while emphasizing how this study seeks to confirm earlier observation at a larger scale.

5. We added additional justification for focusing on mature (>100 years) forest.
6. We added clarification about the forest inventory sampling design and discussion as to why difference in NPP between the field and satellite data sets did not reflect differences in sampling extent.
7. We clarified that computing carbon residence time as the ratio of biomass to productivity assumes that both biomass and productivity are not changing through time. We also added text to the discussion illustrating that the changes in CRT that we observed we not due to differences in stand age, but rather climate.
8. We added a comparison of field- and satellite-derived estimates of NPP, BIO, and CRT.

[revised manuscript text omitted]

(a) Tree BIO$_{field}$ and CMI$_{wy}$
(b) CMI$_{wy}$
(c) Tree NPP$_{sat}$
(d) Tree BIO$_{sat}$
N
45° N
40° N
35° N
-120° W
-115° W
0
250
km
500
CMI$_{wy}$ 1985-2014 (cm yr$^{-1}$)
< -200
-200 - -100
-100 - 0
0 - 100
100 - 200
> 200
NPP$_{sat}$ 2000-2014 (Mg C ha$^{-1}$ yr$^{-1}$)
0.1 - 2.5
2.5 - 5.0
5.0 - 7.5
7.5 - 10.0
10.0 - 24.0
BIO$_{field}$ 1999-2014 (Mg C ha$^{-1}$)
1 - 100
100 - 200
200 - 400
400 - 600
600 - 950
BIO$_{sat}$ c. 2000-2008 (Mg C ha$^{-1}$)
1 - 25
25 - 50
50 - 100
100 - 200
200 - 680

**Figures**

[Figure]

Figure 1. Mean climatic moisture index ($CMI_{\overline{wy}}$; cm yr$^{-1}$), tree net primary productivity (NPP; Mg C ha$^{-1}$ yr$^{-1}$), and live tree biomass (BIO; Mg C ha$^{-1}$) in the western US. (a) BIO derived from field measurements ($BIO_{field}$) at mature sites (>100 years) in WAORCA. For visual clarity only 20% of the 1,953 sites are depicted. (b) $CMI_{\overline{wy}}$ was computed as monthly precipitation minus reference evapotranspiration summed over the annual water year (October-September) and then averaged from 1985-2014. (c) Mean annual NPP was quantified using MODIS satellite data from 2000-2014 ($NPP_{sat}$). (d) BIO was quantified using satellite-derived estimates of carbon stocks ($BIO_{sat}$).

[Figure]

(a) WAORCA
(b) WAORCA
(c) WAORCA
(d) Western US
(e) Western US
(f) Western US
Tree NPPfield (Mg C ha-1 yr-1)
Tree BIOfield (Mg C ha-1)
Tree CRTfield (yr)
Tree NPPsat (Mg C ha-1 yr-1)
Tree BIOsat (Mg C ha-1)
Tree CRTsat (yr)
Average climate moisture index (cm yr-1)

[Figure]

Figure 2. Tree net primary productivity (NPP; Mg C $ha^{-1}$ $yr^{-1}$), live biomass (BIO; Mg C $ha^{-1}$), and carbon residence time (CRT; years) increased with increasing  water availability across both WAORCA (a-c) and the broader western US (d-f). Forest characteristics were derived from field measurements on 1,953 inventory plots in WAORCA (a-c) and from satellite remote sensing data sets across 18 Mha of mature forest in the western US (d-f). $NPP_{sat}$ was characterized using MODIS data averaged annually from 2000 to 2014. $BIO_{sat}$ was quantified based on an ensemble of aboveground biomass maps plus estimates of coarse root, fine root, and foliage biomass. CRT was computed for each field plot and pixel as BIO / NPP. Water availability was quantified using a climate moisture index (CMI= P-$ET_0$) summed over the water year (October-September) and then averaged from 1985-2014 ($CMI_{\overline{wy}}$). The region was partitioned into 10 cm $yr^{-1}$ (non-overlapping) $CMI_{\overline{wy}}$ bins, pixels/plots were allocated to

bins, and then forest characteristics were summarized within each bin. In each panel, the bold line denotes the median, dark gray band the 25-75$^{th}$ percentiles, and light gray band the 10-90$^{th}$ percentiles. Note the different y-axis scales between (b) and (e), as well as (c) and (f).